# Purify Perturbative Availability Poisoning Attacks via Rate-Constrained Variational Autoencoders

## Abstract

Perturbative availability poisoning attacks seek to maximize testing error by making subtle modifications to training examples that are correctly labeled. Defensive strategies against these attacks can be categorized based on whether specific interventions are adopted during the training phase. The first approach is training-time defense, such as adversarial training, which can effectively mitigate poisoning effects but is computationally intensive. The other approach is pre-training purification, *e.g.,* image short squeezing, which consists of several simple compressions but often encounters challenges in dealing with various poison types. Our work provides a novel disentanglement mechanism to build an efficient pre-training purification method that achieves superior performance to all existing defenses. Firstly, we uncover rate-constrained variational autoencoders (VAEs), demonstrating a clear tendency to suppress poison patterns by minimizing mutual information in the latent space. We subsequently conduct a theoretical analysis to offer an explanation for this phenomenon. Building upon these insights, we introduce a disentangle variational autoencoder (D-VAE), capable of disentangling the added perturbations with learnable class-wise embeddings. Based on this network, a two-stage purification approach is naturally developed. The first stage focuses on roughly suppressing poison patterns, while the second stage produces refined, poison-free results, ensuring effectiveness and robustness across various scenarios and datasets. Extensive experiments demonstrate the remarkable performance of our method across CIFAR-10, CIFAR-100, and a 100-class ImageNet-subset.

## 1 Introduction

Although machine learning (ML) models often achieve impressive performance on a range of challenging tasks, their effectiveness can significantly deteriorate in the presence of the gaps between the training and testing data distributions. One of the most widely studied types of these gaps is related to the vulnerability of standard models to adversarial examples (Goodfellow et al., 2014; Madry et al., 2018), posing a significant threat to the inference phase. However, an even more destructive and often underestimated threat emerges from malicious perturbations during the training phase, namely perturbative availability poisoning attacks, which seek to maximize testing error by making subtle modifications of correctly labeled training examples (Feng et al., 2019).

In the era of big data, vast amounts are freely collected from the Internet, powering remarkable advances in deep neural networks (Schmidhuber, 2015). Nonetheless, it's essential to note that online data may contain proprietary or private information, raising concerns about unauthorized use. Perturbative availability poisoning attacks are considered a promising route to data protection (Huang et al., 2021). Recently, a significant number of efforts have emerged to add those perturbations to images as shortcuts to disrupt the training process (Yu et al., 2022). On the other hand, data exploiters perceive these protection techniques as potential threats to a company's commercial interests, leading to extensive research efforts in developing defenses. Previous research has demonstrated that training-time defenses, such as adversarial training and adversarial augmentations, can alleviate poisoning effects. However, their practicality is limited by the massive computational costs. Recently, preprocessing-based defenses have gained attention with simple compressions like JPEG and grayscale demonstrating the advantages over adversarial training in computational effi-

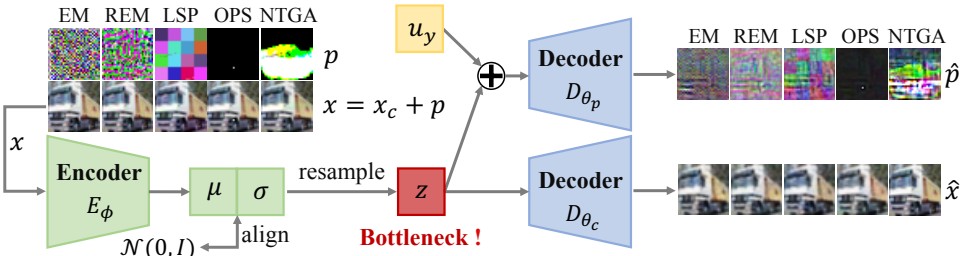

Figure 1: A visual depiction of D-VAE is presented, containing two components. One component generates reconstructed images $\hat{x}$, preserving the primary content of poisoned inputs $x$. The auxiliary decoder maps a trainable class-wise embedding $u_y$ and latents $z$ to disentangled perturbations $\hat{p}$. Here, $x_c$ is clean data, and $p$ denote added perturbations. Perturbations are normalized for clarity.

ciency (Liu et al., 2023). However, these methods lack universality, as different compression techniques might be best suited for various poison types. Pre-training purification has demonstrated great potential in addressing the issue of perturbative availability poisoning attacks in both effectiveness and efficiency (Liu et al., 2023). This kind of method doesn't intervene in the model's training but instead concentrates on refining the data, which well aligns with the recent theme of data-centric AI (DCAI) (Zha et al., 2023). Focusing on fundamental data-related issues rather than relying on untrusted or compromised data leads to more reliable and effective machine learning models.

In this paper, we focus on the pre-training purification paradigm. Our overall approach is to utilize a disentanglement mechanism to separate the poison signal from the intrinsic signal of the image with a rate-constrained VAE to obtain clean data. Firstly, we discover that a rate-constrained VAE can effectively remove poison patterns by minimizing mutual information in latents when compared to JPEG (Guo et al., 2018), with a derived detailed theoretical explanation. Then, we formulate perturbative availability poisoning attacks as the transformation of less-predictive features into highly predictive ones. This perspective reveals that perturbations with a larger inter-class distance and smaller intra-class variance can create stronger poisoning attacks by shifting the optimal separating hyperplane of a Bayes classifier. Subsequently, we show that VAEs are particularly effective at suppressing perturbations possessing these characteristics. Furthermore, we observe that most poison patterns exhibit lower class-conditional entropy. Thus, we propose a method involving class-wise embeddings to disentangle these added perturbations. Building upon these findings, we present a purification framework that offers consistent and adaptable defense against availability poisoning attacks. With this method, models trained on purified datasets can achieve competitive performance, reaching around 91% on CIFAR-10 (Krizhevsky et al., 2009) and 75% on the ImageNet-subset (Deng et al., 2009), even better than the strongest training-time defenses and diffusion model purification that requires extra clean data.

In summary, our contributions can be outlined as follows:

- We discover that rate-constrained VAEs exhibit a preference for removing availability poison patterns, and offer a comprehensive theoretical analysis to support this finding.
- We introduce D-VAE, a network that can disentangle the added perturbations. Our additional evaluations also show that D-VAE can purify the poisoned data within a mixed dataset, and increase the volume of poisoned data while adhering to a small ratio (1%).
- On top of the D-VAE, we propose a unified purification framework for countering various perturbative availability poisoning attacks. Extensive experiments demonstrate the remarkable performance of our method across CIFAR-10, CIFAR-100, and a 100-class ImageNet-subset, encompassing multiple poison types and different perturbation strengths, *e.g.,* with only around 4% drop on ImageNet-subset compared to models trained on clean data.

## 2 RELATED WORK

### 2.1 DATA POISONING

Data poisoning attacks (Barreno et al., 2010; Goldblum et al., 2022), involving the manipulation of training data to disrupt the performance of models during inference, can be broadly categorized

into two main types: integrity attacks and availability attacks. Integrity attacks aim to manipulate the model's output during inference (Barreno et al., 2006; Xiao et al., 2015; Zhao et al., 2017), *i.e.,* backdoor attacks (Gu et al., 2017; Schwarzschild et al., 2021; Yu et al., 2023), where the model behaves maliciously only when presented with data containing specific triggers. In contrast, availability attacks aim to degrade the overall performance on validation and test datasets (Biggio et al., 2012; Xiao et al., 2015). Typically, such attacks inject poisoned data into the clean training set. Poisoned samples are usually generated by adding unbounded perturbations, and take only a fraction of the entire dataset (Koh & Liang, 2017; Zhao & Lao, 2022; Lu et al., 2023). These methods are primarily designed for malicious purposes, and the poisoned samples are relatively distinguishable.

Another recent emerging type is perturbative availability poisoning attacks (Feng et al., 2019; Fowl et al., 2021; Chen et al., 2023; Liu et al., 2023), where samples from the entire training dataset undergo subtle modifications (*e.g.,* bounded perturbations $\|\boldsymbol{p}\|_{\infty} \leq \frac{8}{255}$), and are correctly labeled. This type of attack, also known as unlearnable samples, can be viewed as a promising approach for data protection. Models trained on such datasets often approach random guessing performance on clean test data. Huang et al. (2021) employ error-minimizing noise (EM) as perturbations. Yuan & Wu (2021) generate protective noise using an ensemble of neural networks modeled with neural tangent kernels (NTGA). Fowl et al. (2021) employ targeted adversarial examples (TAP) as poisoned data. Fu et al. (2022) focuses on conducting robust attacks (REM) against adversarial training. Subsequently, Yu et al. (2022) explore effecient and surrogate-free attacks (LSP), and extending the perturbations to be $\ell_2$ bounded. Recently, Wu et al. (2023) introduce one-pixel shortcuts (OPS), which enhances the robustness to adversarial training and strong augmentations.

## 2.2 EXISTING DEFENSES

Defenses against perturbative availability poisoning attacks can be categorized into training-time defenses and pre-training processing. Huang et al. (2021); Liu et al. (2023) demonstrate that these attacks are robust to data augmentations, *e.g.,* Cutout (DeVries & Taylor, 2017), Mixup (Zhang et al., 2018). Tao et al. (2021) find that adversarial training (AT) could mitigate poisoning effects, but it is computationally expensive and cannot fully restore accuracy (Zhang et al., 2019), resulting in a 10% drop on CIFAR-10. Building on the idea of AT, Qin et al. (2023b) employ adversarial augmentations (AA) with improved performance, but it still demands intensive training and does not generalize well to ImageNet-subset. For pre-training defenses, pre-filtering, *e.g.,* gaussian smoothing, mean/median filtering, also show substantial effects but not comparable to AT (Fowl et al., 2021; Liu et al., 2023). Recently, Liu et al. (2023) utilize simple compressions including JPEG compression, grayscale, and bit depth reduction (Wang et al., 2018) to defense, while each technique do not fit all poisoning approaches. Moreover, it is noted that low-quality JPEG compression, while effective for defense, significantly degrades image quality. Dolatabadi et al. (2023) and Jiang et al. (2023) employ a diffusion model for purification, but this method necessitates a substantial amount of additional clean data to train the diffusion model (Ho et al., 2020; Song et al., 2021), making it impractical.

## 3 D-VAE: DISENTANGLE PERTURBATIVE POISON PATTERNS

### 3.1 PRELIMINARIES

Formally, for perturbative availability poisoning attacks, all training data can be perturbed to some extent, while the labels should remain correct (Feng et al., 2019; Fowl et al., 2021). The task to craft poisoning perturbations can be formalized into the following bi-level optimization problem:

$$\max_{\boldsymbol{p} \in \mathcal{S}} \mathbb{E}_{(\boldsymbol{x_c}, y) \sim \mathcal{D}} \big[ \mathcal{L}(F(\boldsymbol{x_c}; \theta^*(\boldsymbol{p})), y) \big] \text{ s.t. } \theta^*(\boldsymbol{p}) = \arg\min_{\theta} \sum_{(\boldsymbol{x_c}^{(i)}, y^{(i)}) \in \mathcal{T}} \mathcal{L}(F(\boldsymbol{x_c}^{(i)} + \boldsymbol{p}^{(i)}; \theta), y^{(i)}), \quad (1)$$

where $\boldsymbol{x_c}$ is the clean data, and $\mathcal{S}$ is the feasible region for perturbations. By adding perturbations $\boldsymbol{p}^{(i)}$ to samples $\boldsymbol{x_c}^{(i)}$ from the clean training dataset $\mathcal{T}$ to formulate the poisoned training dataset $\mathcal{P}$, the adversary aims to induce poor generalization of the trained model $F$ to the clean test dataset $\mathcal{D}$. Conversely, data exploiters aim to obtain the learnable data by employing a mapping $g$ such that:

$$\min_{g} \mathbb{E}_{(\boldsymbol{x_c}, y) \sim \mathcal{D}} \big[ \mathcal{L}(F(\boldsymbol{x_c}; \theta^*(g)), y) \big] \text{ s.t. } \theta^*(g) = \arg\min_{\theta} \sum_{(\boldsymbol{x_c}^{(i)} + \boldsymbol{p}^{(i)}, y^{(i)}) \in \mathcal{P}} \mathcal{L}(F(g(\boldsymbol{x_c}^{(i)} + \boldsymbol{p}^{(i)}); \theta), y^{(i)}), \quad (2)$$

where $\boldsymbol{p}^{(i)}$ is defined in Eq. 1, and $\mathcal{P}$ is the poisoned training data. In this paper, we focus on pre-training purification, where $g$ is applied for that purification, before training the classifier.

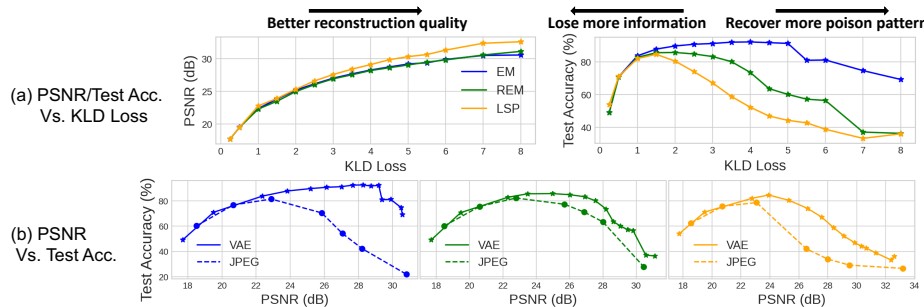

Figure 2: (a): Results using VAEs: PSNR/Test Acc. Vs. KLD Loss is assessed on the poisoned CIFAR-10. (b): Comparison between VAEs and JPEG: PSNR Vs. Test Acc. We adopt JPEG with quality {2,5,10,30,50,70,90}. Specifically, we include EM, REM, and LSP as attack methods here.

## 3.2  A VAE CAN EFFECTIVELY MITIGATE THE IMPACT OF POISON PATTERNS WITH ITS CONSTRAINED REPRESENTATION CAPACITY

The VAE maps the input to a lower-dimensional latent space, generating parameters for a variational distribution. The decoder reconstructs data from this latent space. The loss function combines a reconstruction loss ("distortion") with a Kullback-Leibler (KL) divergence term ("rate"), acting as a limit on mutual information and serving as a compression regularizer (Bozkurt et al., 2021).

Since perturbative availability poisoning attacks have demonstrated vulnerability to compression techniques like JPEG, we first investigate whether a rate-constrained VAE can eliminate these poison patterns and obtain the restored learnable samples. In essence, we introduce an updated loss function incorporating a rate constraint as follows ($\mathcal{P}$ is the poisoned dataset, and $\boldsymbol{x}$ is the poisoned image):

$$\mathcal{L}_{\text{VAE}} = \sum_{\boldsymbol{x},y\in\mathcal{P}} \underbrace{\|\boldsymbol{x}-\hat{\boldsymbol{x}}\|_2^2}_{\text{distortion}} + \lambda \cdot \underbrace{\max(\text{KLD}(\boldsymbol{z},\mathcal{N}(\boldsymbol{0},\boldsymbol{I})),\text{kld}_{\text{limit}})}_{\text{rate constraint}}, \tag{3}$$

where $\boldsymbol{z}$ is the latent feature, $\hat{\boldsymbol{x}}$ is the reconstructed image, the KLD Loss is formulated from Kingma & Welling (2014) and provided in the Appendix B.1, and the kld$_{\text{limit}}$ serves as the upper bound for the KLD loss. Then, we proceed to train the VAE on the poisoned CIFAR-10. Subsequently, we report the accuracy on the clean test set achieved by a ResNet-18 trained on the reconstructed images. In Figure 2(a), reducing the KLD loss decreases reconstruction quality (measured by PSNR between $\hat{\boldsymbol{x}}$ and $\boldsymbol{x}$). This reduction can eliminate added poison patterns and original valuable features. The right image of Figure 2(a) shows that increased removal of poison patterns in $\hat{\boldsymbol{x}}$ correlates with improved test accuracy. However, heavily corrupting $\hat{\boldsymbol{x}}$ by further reducing kld$_{\text{limit}}$ removes more valuable features, leading to a drop in test accuracy. In Figure 2(b), the comparison with JPEG at various quality settings shows that when processed through VAEs and JPEG to achieve similar PSNR, test accuracy with VAEs is higher than JPEG. This suggests that VAEs are significantly more effective at eliminating poison patterns than JPEG compression, when achieving similar levels of reconstruction quality. Then, we delve into the reasons why VAEs can exhibit such preference.

## 3.3  THEORETICAL ANALYSIS AND INTRINSIC CHARACTERISTICS

Given that the feature extractor's function in mapping input data to the latent space is pivotal for the classification process conducted by DNNs, we conduct our analysis on the latent features $\boldsymbol{v}$.

**Hyperplane shift caused by poisoning attacks.** Consider the following binary classification problem with regards to the features extracted from the data $\boldsymbol{v} = (\boldsymbol{v}_c, \boldsymbol{v}_s^t)$ consisting of a predictive feature $\boldsymbol{v}_c$ of a Gaussian mixture $\mathcal{G}_c$ and a non-predictive feature $\boldsymbol{v}_s^t$ which follows:

$$y \overset{u.a.r}{\sim} \{0,1\}, \quad \boldsymbol{v}_c \sim \mathcal{N}(\boldsymbol{\mu}_c^y, \boldsymbol{\Sigma}_c), \quad \boldsymbol{v}_s^t \sim \mathcal{N}(\boldsymbol{\mu}^t, \boldsymbol{\Sigma}^t), \quad \boldsymbol{v}_c \perp\!\!\!\perp \boldsymbol{v}_s^t, \quad \Pr(y=0)=\Pr(y=1). \tag{4}$$

**Proposition 1** *For the features $\boldsymbol{v} = (\boldsymbol{v}_c, \boldsymbol{v}_s^t)$ following the distribution (4), the optimal separating hyperplane using a Bayes classifier is formulated by:*

$$\boldsymbol{w}_c^T\left(\boldsymbol{v}_c^* - \frac{\boldsymbol{\mu}_c^0 + \boldsymbol{\mu}_c^1}{2}\right) = 0, \quad \text{where } \boldsymbol{w}_c = \boldsymbol{\Sigma}_c^{-1}(\boldsymbol{\mu}_c^0 - \boldsymbol{\mu}_c^1). \tag{5}$$

The proof is provided in Appendix A.1. Subsequently, we assume that a malicious attacker modifies $\boldsymbol{v}_s^t$ to $\boldsymbol{v}_s$ of the following distributions $\mathcal{G}_s$ to make it predictive for training a Bayes classifier:

$$y \overset{u.a.r}{\sim} \{0,1\}, \quad \boldsymbol{v}_s \sim \mathcal{N}(\boldsymbol{\mu}_s^y, \boldsymbol{\Sigma}_s), \quad \boldsymbol{v}_c \perp\!\!\!\perp \boldsymbol{v}_s. \tag{6}$$

**Theorem 1** *Consider features from the training data for the Bayes classifier is modified from $\boldsymbol{v} = (\boldsymbol{v}_c, \boldsymbol{v}_s^t)$ in Eq. 4 to $\boldsymbol{v} = (\boldsymbol{v}_c, \boldsymbol{v}_s)$ in Eq. 6, the hyperplane is shifted with a distance given by:*

$$d = \frac{\|\boldsymbol{w}_s^T(\boldsymbol{v}_s - \frac{\boldsymbol{\mu}_s^0 + \boldsymbol{\mu}_s^1}{2})\|_2}{\|\boldsymbol{w}_c\|_2}, \quad \text{where } \boldsymbol{w}_c = \boldsymbol{\Sigma}_c^{-1}(\boldsymbol{\mu}_c^0 - \boldsymbol{\mu}_c^1), \; \boldsymbol{w}_s = \boldsymbol{\Sigma}_s^{-1}(\boldsymbol{\mu}_s^0 - \boldsymbol{\mu}_s^1). \quad (7)$$

The proof is provided in Appendix A.2. When conducting evaluations on the testing data that follows the same distribution as the clean data $\boldsymbol{v} = (\boldsymbol{v}_c, \boldsymbol{v}_s^t)$, with the term $\boldsymbol{v}_s$ in Eq. 7 replaced by $\boldsymbol{v}_s^t$, it leads to a greater prediction error if $\|\boldsymbol{w}_s\|_2 \gg \|\boldsymbol{w}_c\|_2$. Theorem 1 indicates that perturbations which create strong attacks tend to have a larger inter-class distance and a smaller intra-class variance.

**Error when aligning with a normal distribution.** Consider a variable $\boldsymbol{v} = (v_1, \ldots, v_d)$ following a mixture of two Gaussian distributions $\mathcal{G}$:

$$y \overset{u.a.r}{\sim} \{0, 1\}, \quad \boldsymbol{v} \sim \mathcal{N}(\boldsymbol{\mu}^y, \boldsymbol{\Sigma}), \quad v_i \perp\!\!\!\perp v_j, \quad \Pr(y = 0) = \Pr(y = 1),$$
$$v_i \sim \mathcal{N}(\mu_i^y, \sigma_i), \quad p_{v_i}(v) = [\mathcal{N}(v; \mu_i^0, \sigma_i) + \mathcal{N}(v; \mu_i^1, \sigma_i)]/2. \quad (8)$$

Each dimensional feature $v_i$ is also modeled as a Gaussian mixture. To start, we normalize each feature through a linear operation to achieve a distribution with zero mean and unit variance. The linear operation and the modified density function can be expressed as follows:

$$z_i = \frac{v_i - \hat{\mu}_i}{\sqrt{(\sigma_i)^2 + (\delta_i)^2}}, \; p_{z_i}(v) = \frac{p_0(v) + p_1(v)}{2}, \; p_0(v) = \mathcal{N}(v; -\hat{\delta}_i, \hat{\sigma}_i), \; p_1(v) = \mathcal{N}(v; \hat{\delta}_i, \hat{\sigma}_i)$$
$$(9)$$
$$\text{where } \hat{\mu}_i = \frac{\mu_i^0 + \mu_i^1}{2}, \; \delta_i = |\frac{\mu_i^0 - \mu_i^1}{2}|, \; \hat{\delta}_i = \delta_i / \sqrt{(\sigma_i)^2 + (\delta_i)^2}, \; \hat{\sigma}_i = \sigma_i / \sqrt{(\sigma_i)^2 + (\delta_i)^2}.$$

**Theorem 2** *Denote $r_i = \frac{\delta_i}{\sigma_i} > 0$, the Kullback–Leibler divergence between $p_{z_i}(v)$ in (9) and a standard normal distribution $\mathcal{N}(v; 0, 1)$ is bounded by:*

$$\frac{1}{2} \ln\left(1 + (r_i)^2\right) - \ln 2 \le KLD(p_{z_i}(v) \| \mathcal{N}(v; 0, 1)) \le \frac{1}{2} \ln\left(1 + (r_i)^2\right), \quad (10)$$

*and observes the following property:*

$$\uparrow r_i \quad \implies \quad \uparrow S(r_i) = KLD(p_{z_i}(v) \| \mathcal{N}(v; 0, 1)). \quad (11)$$

The proof for Eq. 9 and Theroem 2 is provided in Appendix A.3.

**Remark 1** *Consequently, if we aim to estimate a normalized Gaussian mixture distribution $z_i \sim p_{z_i}(v)$ using $\widehat{P}$ subject to $KLD(\widehat{P} \| \mathcal{N}(0, 1)) < \epsilon$. Then for features $v_i \in \{V | r_V < S^{-1}(\epsilon)\}$, we can employ an identical mapping $\widehat{P} = p_{z_i}(v)$ to estimate the distributions of $z_i$, resulting in zero estimation error. However, for features $v_i \notin \{V | r_V < S^{-1}(\epsilon)\}$, an estimation error, denoted as $\int_{-\infty}^{\infty} [\widehat{P}(v) - p_{z_i}(v)]^2 dv$, is inevitable, and is proportional to $(r_{v_i} - S^{-1}(\epsilon))$. And the estimated $\widehat{P}$ is constrained to have a smaller $r$, making it less predictive for classification.*

Remark 1 indicates that perturbative patterns that make strong poisoning attacks tend to suffer from larger errors when estimating with distributions subject to the constraint on the KLD. Thus, the training of a rate-constrained VAE includes simulating the process of mapping the data to latent representations and aligning them with a normal distribution to a certain extent. The decoder learns to reconstruct the input data from the resampled latents $\boldsymbol{z}$. Consequently, the highly predictive shortcuts are subdued or eliminated in the reconstructed data $\hat{\boldsymbol{x}}$.

**Proposition 2** *The conditional entropy of a Gaussian mixture $\boldsymbol{v}_s$ of $\mathcal{G}_s$ in Eq. 6 is given by:*

$$H(\boldsymbol{v}_s | y_i) = \frac{dim(\boldsymbol{v}_s)}{2}(1 + \ln(2\pi)) + \frac{1}{2} \ln |\boldsymbol{\Sigma}_s|, \quad (12)$$

*where $dim(\boldsymbol{v}_s)$ is the dimensions of the features. If each feature $v_s^d$ is independent, then:*

$$H(\boldsymbol{v}_s | y_i) = \frac{dim(\boldsymbol{v}_s)}{2}(1 + \ln(2\pi)) + \sum_{d=1}^{dim(\boldsymbol{v}_s)} \ln \sigma_s^d. \quad (13)$$

As the inter-class distance $\Delta_s = \|\boldsymbol{\mu}_s^0 - \boldsymbol{\mu}_s^1\|_2$ is constrained to ensure the invisibility of the poison patterns, most availability poison patterns exhibit a relatively low intra-class variance. Proposition 2 suggests that the class-conditional entropy of the perturbations is comparatively low. Adversarial poisoning (Fowl et al., 2021) could be an exception since they can maximize latent space shifts with minimal perturbation in the RGB space. However, the preference to be removed by VAE still holds.

---

**Algorithm 1** Purification of poisoned samples with D-VAE

---

1: **Input:** poisoned dataset $\mathcal{P}^0$, D-VAE ($E_\phi$, $D_{\theta_c}$, $D_{\theta_p}$, $\boldsymbol{u_y}$), kld$_{\text{limit}}$: $kld_1$, $kld_2$
2: # First stage: recover and remove heavy poison patterns by training D-VAE with small $kld_1$
3: Randomly initialize ($\phi$, $\theta_c$, $\theta_p$, $\boldsymbol{u_y}$), and using Adam to minimize Eq. 14 on $\mathcal{P}^0$ with $kld_1$
4: Inference with trained VAE on $\mathcal{P}^0$, and save a new dataset $\mathcal{P}^1$ with sample $\boldsymbol{x}^1 = \boldsymbol{x}^0 - \hat{\boldsymbol{p}}^0$
5: # Second stage: generate purified data by training D-VAE with larger $kld_2$
6: Randomly initialize ($\phi$, $\theta_c$, $\theta_p$, $\boldsymbol{u_y}$), and using Adam to minimize Eq. 14 on $\mathcal{P}^1$ with $kld_2$
7: Inference with trained VAE on $\mathcal{P}^1$, and save a new dataset $\mathcal{P}^2$ with sample $\boldsymbol{x}^2 = \boldsymbol{x}^1 - \hat{\boldsymbol{p}}^1$
8: Inference with trained VAE on $\mathcal{P}^2$, and save a new dataset $\mathcal{P}^3$ with sample $\boldsymbol{x}^3 = \hat{\boldsymbol{x}}^2$
9: **Return** purified dataset $\mathcal{P}^3$

---

### 3.4 D-VAE: Rate-constrained VAE with poison disentanglement

Given that the poison patterns for each class exhibit low entropy, this implies that they can primarily be reconstructed using representations with limited capacity. Since poison patterns are typically crafted to be sample-wise, we propose to integrate trainable class-wise embeddings $\boldsymbol{u_y}$ and latents $\boldsymbol{z}$ into the decoding network to facilitate the reconstruction of these poison patterns. The overall network is in Figure 1, and the improved loss to optimize the D-VAE ($E_\phi$, $D_{\theta_c}$, $D_{\theta_p}$, $\boldsymbol{u_y}$) is given:

$$\mathcal{L}_{\text{D-VAE}} = \sum_{\boldsymbol{x},\boldsymbol{y}\in\mathcal{P}} \underbrace{\|\boldsymbol{x} - \hat{\boldsymbol{x}}\|_2^2}_{\text{distortion}} + \underbrace{\|(\boldsymbol{x} - \hat{\boldsymbol{x}}) - \hat{\boldsymbol{p}}\|_2^2}_{\text{recover poison patterns}} + \lambda \cdot \underbrace{\max(\text{KLD}(\boldsymbol{z}, \mathcal{N}(\boldsymbol{0}, \boldsymbol{I})), \text{kld}_{\text{limit}})}_{\text{rate constraint}}, \quad (14)$$

where $\boldsymbol{\mu}, \boldsymbol{\sigma} = E_\phi(\boldsymbol{x})$, $\boldsymbol{z}$ is sampled from $\mathcal{N}(\boldsymbol{\mu}, \boldsymbol{\sigma})$, $\hat{\boldsymbol{x}} = D_{\theta_c}(\boldsymbol{z})$, disentangled poison patterns $\hat{\boldsymbol{p}} = D_{\theta_p}(\boldsymbol{u_y} + \boldsymbol{z})$. Similar to training VAEs using Eq. 3, minimizing $\|\boldsymbol{x} - \hat{\boldsymbol{x}}\|_2^2$ in Eq. 14 also compels the decoder to fully utilize the latents $\boldsymbol{z}$ for reconstruction. However, when setting a low upper limit on the KLD loss, $\hat{\boldsymbol{x}}$ cannot be perfectly reconstructed and contains very few poison patterns, resulting in a significant portion of poison patterns $\boldsymbol{p}$ remaining in the residuals $\boldsymbol{x} - \hat{\boldsymbol{x}}$. Inspired by this, aligning $\boldsymbol{x} - \hat{\boldsymbol{x}}$ with $\hat{\boldsymbol{p}}$ allows the network to be able to map $\boldsymbol{u_y} + \boldsymbol{z}$ to the poison patterns.

### 3.5 Purify poisoned samples with D-VAE

Given that a large KLD limit fails to effectively surpress the poisoning attacks, while a small one might significantly deteriorate the quality of reconstructed images, we introduce a two-stage purification framework as shown in Algorithm 1. In the first stage, we use a small kld$_{\text{limit}}$ to train the VAE with the poisoned dataset $\mathcal{P}^0$. This approach allows us to reconstruct a significant portion of the poison patterns. During inference, we subtract the input $\boldsymbol{x}^0$ from $\mathcal{P}^0$ by the predicted poison patterns $\hat{\boldsymbol{p}}^0$ and obtain these modified images as $\mathcal{P}^1$. In the second stage, we set a larger kld$_{\text{limit}}$ for training. After subtracting $\boldsymbol{x}^1$ by $\hat{\boldsymbol{p}}^1$ and saving it as $\boldsymbol{x}^2$ in the first inference. Since the poison patterns are learned in an unsupervised manner, it is challenging to achieve complete reconstruction. Hence, we proceed with a second inference and obtain the output $\hat{\boldsymbol{x}}^2$ as the final result.

## 4 Experiments

### 4.1 Experimental Setup

**Datasets and models.** We choose three commonly used datasets: CIFAR-10, CIFAR-100 (Krizhevsky et al., 2009), and a subset of ImageNet (Deng et al., 2009) comprising the first 100 classes. For CIFAR-10 and CIFAR-100, we maintain the original size of $32 \times 32$. Regarding the ImageNet subset, we follow prior research (Huang et al., 2021), and resize the image to $224 \times 224$. In our main experiments, we adopt the ResNet-18 (He et al., 2016) model as both the surrogate and target model. To evaluate transferability, we include various classifiers, such as ResNet-50 (RN-50), DenseNet-121 (DN-121) (Huang et al., 2017), MobileNet-V2 (MN-v2) (Sandler et al., 2018).

**Perturbative availability poisoning attacks.** We examine several representative attacking methods with various perturbation bounds. The majority of methods rely on a surrogate model, including NTGA (Yuan & Wu, 2021), EM (Huang et al., 2021), REM (Fu et al., 2022), TAP (Fowl et al., 2021), SEP (Chen et al., 2023), and employ the $\ell_\infty$ bound. On the other hand, surrogate-free methods such as LSP (Yu et al., 2022) and AR (Sandoval-Segura et al., 2022) utilize the $\ell_2$ bound. Additionally,

OPS (Wu et al., 2023) utilizes the $\ell_0$ bound. The diversity of these attacking methods can validate the generalization capacity of our proposed purification framework.

**Competing defensive methods.** We include two training-time defenses: adversarial training (AT) with $\epsilon = 8/255$ (Wen et al., 2023) and adversarial augmentations (AA) (Qin et al., 2023b). Among the pre-training methods, we include ISS (Liu et al., 2023), consisting of bit depth reduction (BDR), Grayscale, and JPEG, as well as AVATAR (Dolatabadi et al., 2023) (denoted as AVA.), which employs a diffusion model trained on the clean CIFAR-10 dataset to purify. For a fair comparison, we choose to report the test accuracy from the last epoch. More details are in the Appendix C.2.

**Model Training.** To ensure consistent training procedures for the classification model, we have formalized the standard training approach. For CIFAR-10, we use 60 epochs, while for CIFAR-100 and the ImageNet subset, 100 epochs are allowed. In all experiments, we use SGD optimizer with an initial learning rate of 0.1 and the CosineAnnealingLR scheduler, keeping a consistent batch size of 128. For D-VAE training on poisoned CIFAR-10/100 dataset, we use a KLD limit of 1.0 in the first stage and 3.0 in the second stage, with only a single $\times 0.5$ downsampling to preserve image quality. For ImageNet, which has higher-resolution images, we employ more substantial downsampling ($\times 0.125$) in the first stage and set a KLD limit of 1.5, while the second stage remains the same as with CIFAR. When comparing the poisoned input and the reconstructed output, these hyperparameters yield PSNRs of around 28 for CIFAR and 30 for ImageNet.

## 4.2 VALIDATE THE EFFECTIVENESS OF THE DISENTANGLED POISON PATTERNS

The availability poisoning attack can be analyzed from the standpoint of shortcut learning (Yu et al., 2022). It has been empirically shown that DNNs trained on the poisoned training data have a tendency to memorize a substantial portion of the perturbations, and subsequently attain high accuracy when applied to data that shares these same perturbations (Liu et al., 2023).

Table 1: Testing accuracy (%) of models trained on reconstructed poisoned dataset $\widehat{\mathcal{P}}$.

| Datasets | Test Set | EM | REM | NTGA | LSP | AR | OPS |
|---|---|---|---|---|---|---|---|
| CIFAR-10 | $\mathcal{T}$ | 9.71 | 19.84 | 29.20 | 15.16 | 13.09 | 18.54 |
| | $\mathcal{D}$ | 9.65 | 19.51 | 28.66 | 15.34 | 12.96 | 18.73 |
| | $\mathcal{P}$ | 91.35 | 99.96 | 99.91 | 99.95 | 100.0 | 99.72 |
| CIFAR-100 | $\mathcal{T}$ | 1.41 | 6.45 | - | 4.24 | 1.63 | 11.29 |
| | $\mathcal{D}$ | 1.39 | 7.69 | - | 4.03 | 1.68 | 10.77 |
| | $\mathcal{P}$ | 98.88 | 96.47 | - | 99.12 | 100.0 | 99.59 |

In this section, we aim to illustrate that the disentangled perturbations remain effective as potent attacks and can be regarded as equivalent to the original poisoned data $\mathcal{P}$. Initially, we look into the amplitude of the perturbations in terms of $\ell_2$-norm. The amplitude of groundtruth $\boldsymbol{p}$ is around 1.0 for LSP and AR, and about 1.5 for others. The generated $\hat{\boldsymbol{p}}$ has an amplitude of about 1.8 for OPS and around 0.7 to 1.0 for others. Notably, the amplitude of $\hat{\boldsymbol{p}}$ is comparable to that of $\boldsymbol{p}$, with $\hat{\boldsymbol{p}}$ being slightly smaller than $\boldsymbol{p}$ except for OPS.

Subsequently, we construct a new poisoned dataset denoted as $\widehat{\mathcal{P}}$ by incorporating the disentangled perturbations $\hat{\boldsymbol{p}}$ into the clean training set $\mathcal{T}$. We proceed to train a model using $\widehat{\mathcal{P}}$, and subsequently evaluate its performance on three distinct sets: the clean training set $\mathcal{T}$, the clean testing set $\mathcal{D}$, and the original poisoned dataset $\mathcal{P}$. From the results in Table 1, it becomes apparent that the reconstructed dataset continues to significantly degrade the accuracy on clean data. In fact, compared to the attacking performance of $\mathcal{P}$ in Table 2 and Table 3, $\widehat{\mathcal{P}}$ even manages to achieve an even superior attacking performance in most instances with less amplitude. During testing on the original poisoned dataset, the accuracy levels are notably high, often approaching 100%. This outcome serves as an indicator of the effectiveness of the disentanglement process.

## 4.3 EXPERIMENTAL RESULTS ON THE PURIFIED DATA

**CIFAR-10 poison purification.** To evaluate the effectiveness of our purification framework, we conducted initial experiments on CIFAR-10. As shown in Table 2, our method consistently provides comprehensive protection against perturbative availability poisoning attacks with varying perturbation bounds and attack methods. In contrast, ISS relies on multiple simple compression techniques and requires adaptive selection of these methods, resulting in subpar defense performance. Notably, when compared to adversarial training, our method achieved an approximately 6% improvement in performance. Even compared with AVATAR, which utilizes a diffusion model trained on the clean CIFAR-10 data, our methods achieve superior performance across all attack methods. Our methods excel, especially on OPS attacks, which often perturb a pixel to its maximum value, creating a robust

Table 2: Clean test accuracy (%) of models trained on the poisoned CIFAR-10 dataset and with our proposed method Vs. other defenses. Our results on additional classifiers are at the rightmost.

| Norm | Attacks/Countermeasures | w/o | AT | AA | BDR | Gray | JPEG | AVA. | Ours | RN-50 | DN-121 | MN-v2 |
|---|---|---|---|---|---|---|---|---|---|---|---|---|
| | Clean (no poison) | **94.57** | 85.17 | 92.27 | 88.95 | 92.74 | 85.47 | 89.61 | **93.29** | 93.08 | 93.73 | 83.61 |
| $\ell_\infty = \frac{8}{255}$ | NTGA (Yuan & Wu, 2021) | 11.10 | 83.63 | 77.92 | 57.80 | 65.26 | 78.97 | 80.72 | **89.21** | 88.96 | 89.28 | 78.72 |
| | EM (Huang et al., 2021) | 12.26 | 84.43 | 67.11 | 81.91 | 19.50 | 85.61 | 89.54 | **91.42** | 91.62 | 91.64 | 81.10 |
| | TAP (Fowl et al., 2021) | 25.44 | 83.89 | 55.84 | 80.18 | 21.50 | 84.99 | 89.13 | **90.48** | 90.50 | 90.51 | 81.28 |
| | REM (Fu et al., 2022) | 22.43 | 86.01 | 64.99 | 32.36 | 62.35 | 84.40 | 86.06 | **86.38** | 85.91 | 86.74 | 79.27 |
| | SEP (Chen et al., 2023) | 6.63 | 83.48 | 61.07 | 81.21 | 8.47 | 84.97 | 89.56 | **90.74** | 90.86 | 90.76 | 80.98 |
| $\ell_2 = 1.0$ | LSP (Yu et al., 2022) | 13.14 | 84.56 | 80.39 | 40.25 | 73.63 | 79.91 | 81.15 | **91.20** | 90.15 | 91.10 | 80.26 |
| | AR (Sandoval-Segura et al., 2022) | 12.50 | 82.01 | 49.14 | 29.14 | 36.18 | 84.97 | 89.64 | **91.77** | 90.53 | 90.99 | 82.26 |
| $\ell_0 = 1$ | OPS (Wu et al., 2023) | 22.03 | 9.48 | 64.02 | 19.58 | 19.43 | 77.33 | 71.62 | **88.95** | 88.10 | 88.78 | 81.40 |

Table 3: Test acc. (%) of models trained on poisoned CIFAR-100.

| Attacks | w/o | AT | AA | ISS | AVA. | Ours |
|---|---|---|---|---|---|---|
| Clean | **77.61** | 59.65 | 69.09 | 71.59 | 61.09 | 70.72 |
| EM | 12.30 | 59.07 | 42.89 | 61.91 | 61.09 | **68.79** |
| TAP | 13.44 | 57.91 | 35.10 | 57.33 | 60.47 | **65.54** |
| REM | 16.80 | 59.34 | 50.12 | 58.13 | 60.90 | **68.52** |
| SEP | 4.66 | 57.93 | 27.77 | 57.76 | 59.80 | **64.02** |
| LSP | 2.91 | 58.93 | 53.28 | 53.06 | 52.17 | **67.73** |
| AR | 2.71 | 58.77 | 26.77 | 56.60 | 60.33 | **63.73** |
| OPS | 12.56 | 7.28 | 36.78 | 54.45 | 44.24 | **65.10** |

Table 4: Test acc. (%) of models trained on poisoned ImageNet-subset.

| Attacks | w/o | AT | AA | ISS | Ours |
|---|---|---|---|---|---|
| Clean | **80.52** | 55.94 | 71.56 | 76.92 | 76.78 |
| EM | 1.08 | 56.74 | 3.82 | 72.44 | **74.80** |
| TAP | 12.56 | 55.36 | 71.38 | 73.24 | **76.56** |
| REM | 2.54 | 59.34 | 20.92 | 58.13 | **72.56** |
| LSP | 2.50 | 58.93 | 46.58 | 53.06 | **76.06** |

Table 5: Performance on poisoned CIFAR-10 with larger bounds: $\ell_\infty = \frac{16}{255}$ and $\ell_2 = 2.0$.

| Attacks | w/o | AT | AA | ISS | AVA. | Ours |
|---|---|---|---|---|---|---|
| EM | 10.09 | 84.02 | 49.23 | 83.62 | 85.61 | **91.06** |
| TAP | 18.45 | 83.46 | 52.92 | 84.98 | 89.43 | **90.55** |
| REM | 23.22 | 35.41 | 50.92 | 75.50 | 52.26 | **79.18** |
| SEP | 12.05 | 83.98 | 56.71 | 85.00 | 88.96 | **90.93** |
| LSP | 15.45 | 79.10 | 59.10 | 41.41 | 41.70 | **86.43** |

shortcut that evades most defenses. Our approach can effectively disentangle the majority of these additive perturbations in the first stage. The subsequent subtraction process can significantly mitigate the poisoning attacks, resulting in the poison-free data in the second stage. The performance of training different classification models on our purified data is reported in the rightmost column of Table 2. As can be observed, our method indeed restores the learnability of data samples.

**CIFAR-100/ImageNet-subset poison purification.** We then expand our experiments to include CIFAR-100 and a 100-class ImageNet subset. Due to the resource-intensive nature of the experiments, we focused on four representative attack methods for the ImageNet subset. Note that for ISS, we report the best accuracy among three compressions. The results, as presented in Table 3 and Table 4, re-confirm the overall effectiveness of our purification framework.

**Experiments on larger perturbations.** In our supplementary experiments, we introduced poisoned samples with larger perturbation bounds. The outcomes on CIFAR-10 are outlined in Table 5. It is worth noting that our method exhibits a high degree of consistency, with almost no performance degradation on EM, TAP, and SEP, and only a slight decrease on REM and LSP. However, it proves to be a challenging scenario for the competing methods to effectively address.

## 4.4 ABLATION STUDY

In this section, we conduct an ablation study on our two-stage purification framework. As shown in Table 6, it becomes evident that the subtraction process in the first stage plays a critical role in mitigating certain attacks, including NTGA, LSP, and OPS. This is particularly evident for LSP, which introduces smooth colorized blocks, and OPS, which perturbs a single pixel to a maximum value, making them challenging to remove when subjected to a moderate KLD limit.

Table 6: Ablation study on the proposed two-stage purification framework on several attacks. s1/s2 denote the first stage and the second stage. $\ominus$ denotes the subtraction operation.

| Attacks | NTGA | EM | TAP | REM | SEP | LSP | AR | OPS | Mean |
|---|---|---|---|---|---|---|---|---|---|
| w/o s1 | 78.62 | **91.85** | **90.97** | 82.06 | 90.76 | 66.76 | 91.39 | 51.71 | 80.52 |
| w/o $\ominus$ in s2 | 87.44 | 91.18 | 90.70 | 85.21 | **90.79** | 90.63 | 91.31 | 84.92 | 89.02 |
| Ours | **89.21** | 91.42 | 90.48 | **86.38** | 90.74 | **91.20** | **91.77** | **88.95** | **90.01** |

## 5 DISCUSSION

### 5.1 PARTIAL POISONING AND UNSUPERVISED POISONED DATA DETECTION

In practical scenarios, it is often the case that only a fraction of the training data can be contaminated with poison. Therefore, in line with previous research, we evaluate these partial poisoning scenarios

Table 7: Performance of detecting poisoned data or increasing poisoned data with various poison ratios on CIFAR-10.

| Task | | Detecting poisoned data | | | | Increasing poisoned data | |
|---|---|---|---|---|---|---|---|
| Attacks | Ratio | Acc. | Recall | Precision | F1-score | Ratio | Test Acc. |
| EM | | 0.918 | 1.0 | 0.709 | 0.830 | | 0.1009 |
| REM | 0.2 | 0.561 | 1.0 | 0.312 | 0.476 | 0.01 | 0.2900 |
| LSP | | 0.777 | 1.0 | 0.472 | 0.641 | | 0.1558 |
| OPS | | 0.724 | 0.993 | 0.420 | 0.590 | | 0.2059 |
| EM | | 0.939 | 1.0 | 0.869 | 0.930 | | 0.1011 |
| REM | 0.4 | 0.785 | 1.0 | 0.651 | 0.789 | 0.02 | 0.2777 |
| LSP | | 0.905 | 1.0 | 0.807 | 0.893 | | 0.1633 |
| OPS | | 0.842 | 0.991 | 0.719 | 0.833 | | 0.2015 |
| EM | | 0.961 | 1.0 | 0.938 | 0.968 | | 0.1229 |
| REM | 0.6 | 0.909 | 0.999 | 0.868 | 0.930 | 0.04 | 0.2319 |
| LSP | | 0.941 | 0.999 | 0.912 | 0.954 | | 0.1405 |
| OPS | | 0.910 | 0.993 | 0.874 | 0.930 | | 0.1632 |
| EM | | 0.982 | 1.0 | 0.978 | 0.989 | | 0.1001 |
| REM | 0.8 | 0.958 | 0.998 | 0.951 | 0.975 | 0.08 | 0.2433 |
| LSP | | 0.973 | 1.0 | 0.968 | 0.984 | | 0.1763 |
| OPS | | 0.932 | 0.997 | 0.924 | 0.959 | | 0.1701 |

Table 8: Clean testing accuracy (%) of models trained on the poisoned CIFAR-10 dataset with different poisoning rate.

| Attacks | Counter | 0.2 | 0.4 | 0.6 | 0.8 |
|---|---|---|---|---|---|
| EM | JPEG | 85.03 | 85.31 | 85.40 | 85.31 |
| | Ours | **93.50** | **93.03** | **93.02** | **92.26** |
| TAP | JPEG | 85.12 | 85.60 | 84.92 | 85.34 |
| | Ours | **90.55** | **90.78** | **90.93** | **91.10** |
| REM | JPEG | 84.64 | 84.90 | 84.62 | 84.97 |
| | Ours | **92.24** | **92.51** | **92.23** | **90.86** |
| SEP | JPEG | 85.34 | 85.22 | 85.06 | 85.06 |
| | Ours | **90.86** | **90.63** | **91.04** | **91.79** |
| LSP | JPEG | 85.22 | 85.34 | 84.26 | 83.02 |
| | Ours | **93.20** | **92.85** | **92.16** | **92.16** |
| AR | JPEG | 85.31 | 85.29 | 85.33 | 84.87 |
| | Ours | **92.77** | **91.83** | **91.41** | **91.70** |
| OPS | JPEG | 85.12 | 84.89 | 84.43 | 83.01 |
| | Ours | **93.15** | **93.29** | **92.13** | **92.16** |

by introducing poison to a specific portion of the training data and subsequently combining it with the remaining clean data for training the target model. We do experiments on CIFAR-10 dataset.

When examining the first stage as outlined in Algorithm 1, we observe that even the perturbations learned for the clean samples can potentially serve as poison patterns. This could be caused by the constrained representation capacity of the class-wise embedding. In essence, building upon this discovery, we have the capability to create a new poisoned dataset denoted as $\widehat{\mathcal{P}}^0$, where each sample is formed as $x^0 + \hat{p}^0$. Models trained on $\widehat{\mathcal{P}}^0$ tend to achieve high prediction accuracy on the poisoned samples but perform notably worse on the clean ones. Consequently, we can employ this metric as a means to detect the presence of poisoned data, and the detection performance is outlined in Table 7. Notably, our detection method attains high accuracy, with an almost 100% recall rate. Subsequently, to address the issue of partial poisoning in datasets, we can adopt a detection-purification approach. The performances of models trained on the purified data are presented in Table 8.

## 5.2 INCREASING POISONED DATA

In this section, we investigate whether our proposed disentanglement approach can help increase the amount of poisoned data when the attacker acquires additional clean data. We conduct experiments on CIFAR-10 dataset by generating poisoned data, denoted as $\mathcal{P}_{(0)}$, using a small ratio of the dataset, while leaving the remaining $\mathcal{T}_{(1)}$ untouched. Subsequently, after training the D-VAE on $\mathcal{P}_{(0)}$, we conduct inference on the clean data $\mathcal{T}_{(1)}$. The addition of $\hat{p}_{(1)}$ to the clean data in $\mathcal{T}_{(1)}$ resulted in a poisoned dataset $\mathcal{P}_{(1)}$. By combining $\mathcal{P}_{(0)}$ and $\mathcal{P}_{(1)}$ to create $\mathcal{P}$, we proceed to train a classifier. The performance on the clean test set $\mathcal{D}$ are reported in Table 7. It is evident that training D-VAE with just 1% poisoned data is adequate for generating additional poisoned samples.

## 6 CONCLUSION

In this paper, we initially demonstrate that a rate-constrained VAE shows its natural preference for removing poison patterns by minimizing mutual information in the latent space. We further provide a theoretical explanation for this behavior. Additionally, our investigations reveal that most perturbative availability poison patterns have a lower class-conditional entropy, and can be disentangled by a learnable class-wise embeddings and a decoding network. Building on these insights, we propose a purification framework that offers a consistent defense against perturbative availability poisoning attacks. Our extensive experiments show that, the remarkable performance of our method across CIFAR-10, CIFAR-100, and a 100-class ImageNet-subset, with different poison types and varying perturbation levels, *i.e.,* only around 4% drop on ImageNet-subset compared to models trained on clean data. We plan to extend our research to purify perturbative availability poisoning attacks that target unsupervised learning scenarios in our future work.

## ETHICS STATEMENT

In summary, our paper presents a effective defense strategy against perturbative availability poisoning attacks, which aim to undermine the overall performance on validation and test datasets by introducing imperceptible perturbations to training examples with accurate labels. Perturbative availability poisoning attacks are viewed as a promising avenue for data protection, particularly to thwart unauthorized use of data that may contain proprietary or sensitive information. However, these protective methods pose challenges to data exploiters who may interpret them as potential threats to a company's commercial interests. Consequently, our method can be employed for both positive usage, such as neutralizing malicious data within a training set, and negative purpose, including thwarting attempts at preserving data privacy. Our proposed method not only serves as a powerful defense mechanism but also holds the potential to be a benchmark for evaluating existing attack methods. We believe that our paper contributes to raising awareness about the vulnerability of current data protection techniques employing perturbative availability poisoning attacks. This, in turn, should stimulate further research towards developing more reliable and trustworthy data protection techniques.

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

# A  PROOFS

In this section, we provide the proofs of our theoretical results in Section 3.3.

## A.1  PROOF OF PROPOSITION 1

Consider the following binary classification problem with regards to the features extracted from the data $\boldsymbol{v} = (\boldsymbol{v}_c, \boldsymbol{v}_s^t)$ consisting of a predictive feature $x_c$ of a Gaussian mixture $\mathcal{G}_c$ and a non-predictive feature $\boldsymbol{v}_s^t$ which follows:

$$y \overset{u.a.r}{\sim} \{0,1\}, \ \boldsymbol{v}_c \sim \mathcal{N}(\boldsymbol{\mu}_c^y, \boldsymbol{\Sigma}_c), \ \boldsymbol{v}_s^t \sim \mathcal{N}(\boldsymbol{\mu}^t, \boldsymbol{\Sigma}^t), \ \boldsymbol{v}_c \perp\!\!\!\perp \boldsymbol{v}_s^t, \quad \Pr(y=0) = \Pr(y=1). \quad (15)$$

**Proposition 1 (restated)** *For the data* $\boldsymbol{v} = (\boldsymbol{v}_c, \boldsymbol{v}_s^t)$ *following the distribution (15), the optimal separating hyperplane using a Bayes classifier is formulated by:*

$$\boldsymbol{w}_c^T(\boldsymbol{v}_c^* - \frac{\boldsymbol{\mu}_c^0 + \boldsymbol{\mu}_c^1}{2}) = 0, \quad \text{where } \boldsymbol{w}_c = \boldsymbol{\Sigma}_c^{-1}(\boldsymbol{\mu}_c^0 - \boldsymbol{\mu}_c^1). \quad (16)$$

*Proof.* Given $\boldsymbol{v} = (\boldsymbol{v}_c, \boldsymbol{v}_s^t)$ following the distribution (15), the optimal decision rule is the maximum a-posteriori probability rule for a Bayes classifier:

$$
\begin{aligned}
i^*(\boldsymbol{v}) &= \arg\max_i \Pr(y=i|\boldsymbol{v}) \\
&= \arg\max_i \left[ \Pr(y=i) \Pr(\boldsymbol{v}|y=i) \right] \\
&= \arg\max_i \left[ \ln \Pr(\boldsymbol{v}|y=i) \right] \\
&= \arg\max_i \left[ \ln \Pr(\boldsymbol{v}_c|y=i) + \ln \Pr(\boldsymbol{v}_s^t|y=i) \right] \\
&= \arg\max_i \left[ \ln \Pr(\boldsymbol{v}_c|y=i) \right] \\
&= \arg\max_i \left[ \ln \left[ (2\pi)^{-\frac{D}{2}} |\boldsymbol{\Sigma}_c|^{-\frac{1}{2}} \exp(-\frac{1}{2}(\boldsymbol{v}_c - \boldsymbol{\mu}_c^i)^\top \boldsymbol{\Sigma}_c^{-1}(\boldsymbol{v}_c - \boldsymbol{\mu}_c^i)) \right] \right] \\
&= \arg\min_i \left[ (\boldsymbol{v}_c - \boldsymbol{\mu}_c^i)^\top \boldsymbol{\Sigma}_c^{-1}(\boldsymbol{v}_c - \boldsymbol{\mu}_c^i) \right] \\
&= \arg\min_i \left[ \boldsymbol{v}_c^\top \boldsymbol{\Sigma}_c^{-1} \boldsymbol{v}_c - 2\boldsymbol{\mu}_c^{i\top} \boldsymbol{\Sigma}_c^{-1} \boldsymbol{v}_c + \boldsymbol{\mu}_c^{i\top} \boldsymbol{\Sigma}_c^{-1} \boldsymbol{\mu}_c^i \right] \\
&= \arg\max_i \left[ \boldsymbol{\mu}_c^{i\top} \boldsymbol{\Sigma}_c^{-1} \boldsymbol{v}_c - \frac{1}{2} \boldsymbol{\mu}_c^{i\top} \boldsymbol{\Sigma}_c^{-1} \boldsymbol{\mu}_c^i \right]
\end{aligned}
\quad , \quad (17)
$$

where $D$ is the dimensions. Thus, the hyperplane is formulated by:

$$
\begin{aligned}
&\boldsymbol{\mu}_c^{0\top} \boldsymbol{\Sigma}_c^{-1} \boldsymbol{v}_c - \frac{1}{2} \boldsymbol{\mu}_c^{0\top} \boldsymbol{\Sigma}_c^{-1} \boldsymbol{\mu}_c^0 = \boldsymbol{\mu}_c^{1\top} \boldsymbol{\Sigma}_c^{-1} \boldsymbol{v}_c - \frac{1}{2} \boldsymbol{\mu}_c^{1\top} \boldsymbol{\Sigma}_c^{-1} \boldsymbol{\mu}_c^1 \\
&\iff \boldsymbol{w}_c^T(\boldsymbol{v}_c^* - \frac{\boldsymbol{\mu}_c^0 + \boldsymbol{\mu}_c^1}{2}) = 0, \quad \text{where } \boldsymbol{w}_c = \boldsymbol{\Sigma}_c^{-1}(\boldsymbol{\mu}_c^0 - \boldsymbol{\mu}_c^1).
\end{aligned}
\quad (18)
$$

## A.2  PROOF OF THEOREM 1

We assume that a malicious attacker modifies $\boldsymbol{v}_s^t$ to $\boldsymbol{v}_s$ of the following distributions $\mathcal{G}_s$ to make it predictive for training a Bayes classifier:

$$y \overset{u.a.r}{\sim} \{0,1\}, \quad \boldsymbol{v}_s \sim \mathcal{N}(\boldsymbol{\mu}_s^y, \boldsymbol{\Sigma}_s), \quad \boldsymbol{v}_c \perp\!\!\!\perp \boldsymbol{v}_s. \quad (19)$$

**Theorem 1 (restated)** *Consider the training data for the Bayes classifier is modified from* $\boldsymbol{v} = (\boldsymbol{v}_c, \boldsymbol{v}_s^t)$ *in Eq. 15 to* $\boldsymbol{v} = (\boldsymbol{v}_c, \boldsymbol{v}_s)$ *in Eq. 19, the hyperplane is shifted with a distance given by*

$$d = \frac{\|\boldsymbol{w}_s^T(\boldsymbol{v}_s - \frac{\boldsymbol{\mu}_s^0 + \boldsymbol{\mu}_s^1}{2})\|_2}{\|\boldsymbol{w}_c\|_2}, \quad \text{where } \boldsymbol{w}_c = \boldsymbol{\Sigma}_c^{-1}(\boldsymbol{\mu}_c^0 - \boldsymbol{\mu}_c^1), \ \boldsymbol{w}_s = \boldsymbol{\Sigma}_s^{-1}(\boldsymbol{\mu}_s^0 - \boldsymbol{\mu}_s^1). \quad (20)$$

*Proof.* After modifying $\boldsymbol{v}_s^t$ to $\boldsymbol{v}_s$, the learned separating hyperplane on the poisoned distributions $\mathcal{G}_p$ = $(\mathcal{G}_c, \mathcal{G}_s)$ turns to (following Proposition 1):

$$w^T\left(\begin{bmatrix} \boldsymbol{v}_c^* - \frac{\boldsymbol{\mu}_c^0 + \boldsymbol{\mu}_c^1}{2} \\ \boldsymbol{v}_s - \frac{\boldsymbol{\mu}_s^0 + \boldsymbol{\mu}_s^1}{2} \end{bmatrix}\right) = 0 \iff \boldsymbol{w}_c^T(\boldsymbol{v}_c^* - \frac{\boldsymbol{\mu}_c^0 + \boldsymbol{\mu}_c^1}{2}) = -\boldsymbol{w}_s^T(\boldsymbol{v}_s - \frac{\boldsymbol{\mu}_s^0 + \boldsymbol{\mu}_s^1}{2}),$$

$$\text{where } \boldsymbol{w} = \begin{bmatrix} \boldsymbol{\Sigma}_c^{-1} & 0 \\ 0 & \boldsymbol{\Sigma}_s^{-1} \end{bmatrix}\begin{bmatrix} \boldsymbol{\mu}_c^0 - \boldsymbol{\mu}_c^1 \\ \boldsymbol{\mu}_s^0 - \boldsymbol{\mu}_s^1 \end{bmatrix} = \begin{bmatrix} \boldsymbol{w}_c \\ \boldsymbol{w}_s \end{bmatrix}, \ \boldsymbol{w}_c = \boldsymbol{\Sigma}_c^{-1}(\boldsymbol{\mu}_c^0 - \boldsymbol{\mu}_c^1). \tag{21}$$

Thus, compared to the original hyperplane as stated in Eq. 16, the hyperplane on the poisoned distribution is shifted with a distance $d$:

$$d = \frac{\|\boldsymbol{w}_s^T(\boldsymbol{v}_s - \frac{\boldsymbol{\mu}_s^0 + \boldsymbol{\mu}_s^1}{2})\|_2}{\|\boldsymbol{w}_c\|_2} \tag{22}$$

When conducting evaluations on the testing data that follows the same distribution as the clean data $\boldsymbol{v} = (\boldsymbol{v}_c, \boldsymbol{v}_s^t)$, with the term $\boldsymbol{v}_s$ in Eq. 22 replaced by $\boldsymbol{v}_s^t$, the shifted distance $d$ is given by

$$d = \frac{\|\boldsymbol{w}_s^T(\boldsymbol{v}_s^t - \frac{\boldsymbol{\mu}_s^0 + \boldsymbol{\mu}_s^1}{2})\|_2}{\|\boldsymbol{w}_c\|_2} \propto \frac{\|\boldsymbol{w}_s\|_2}{\|\boldsymbol{w}_c\|_2}. \tag{23}$$

And it leads to a greater prediction error if $\|\boldsymbol{w}_s\|_2 \gg \|\boldsymbol{w}_c\|_2$.

## A.3  PROOF OF THEOREM 2

Consider a variable $\boldsymbol{v} = (v_1, \ldots, v_d)$ following a mixture of two Gaussian distributions $\mathcal{G}$:

$$y \overset{u.a.r}{\sim} \{0, 1\}, \quad \boldsymbol{v} \sim \mathcal{N}(\boldsymbol{\mu}^y, \boldsymbol{\Sigma}), \quad x_i \perp\!\!\!\perp x_j, \quad \Pr(y=0) = \Pr(y=1),$$

$$v_i \sim \mathcal{N}(\mu_i^y, \sigma_i), \quad p_{v_i}(v) = \frac{\mathcal{N}(v; \mu_i^0, \sigma_i) + \mathcal{N}(v; \mu_i^1, \sigma_i)}{2}. \tag{24}$$

Each dimensional feature $v_i$ is also modeled as a Gaussian mixture. To start, we normalize each feature through a linear operation to achieve a distribution with zero mean and unit variance. Firstly, we calculate the mean and standard deviation of $v_i$:

$$\hat{\mu}_i = \mathbb{E}_{v_i}[v_i] = \frac{\mu_i^0 + \mu_i^1}{2}, \quad \text{Var}[v_i] = \mathbb{E}_{v_i}[(v_i)^2] - \mathbb{E}_{v_i}[v_i]^2 = \sigma_i^2 + (\frac{\mu_i^0 - \mu_i^1}{2})^2. \tag{25}$$

Thus, the linear operation and the modified density function can be expressed as follows:

$$z_i = \frac{v_i - \hat{\mu}_i}{\sqrt{(\sigma_i)^2 + (\delta_i)^2}}, \ p_{z_i}(v) = \frac{p_0(v) + p_1(v)}{2}, \ p_0(v) = \mathcal{N}(v; -\hat{\delta}_i, \hat{\sigma}_i), \ p_1(v) = \mathcal{N}(v; \hat{\delta}_i, \hat{\sigma}_i)$$

$$\text{where } \hat{\mu}_i = \frac{\mu_i^0 + \mu_i^1}{2}, \ \delta_i = |\frac{\mu_i^0 - \mu_i^1}{2}|, \ \hat{\delta}_i = \delta_i/\sqrt{(\sigma_i)^2 + (\delta_i)^2}, \ \hat{\sigma}_i = \sigma_i/\sqrt{(\sigma_i)^2 + (\delta_i)^2}. \tag{26}$$

**Theorem 2 (restated)** *Denote $r = \frac{\delta_i}{\sigma_i} > 0$, the Kullback–Leibler divergence between $p_{z_i}(v)$ and a standard normal distribution $\mathcal{N}(v; 0, 1)$ is tightly bounded by*

$$\frac{1}{2}\ln(1 + r^2) - \ln 2 \le D_{KL}(p_{z_i}(v) \| \mathcal{N}(v; 0, 1)) \le \frac{1}{2}\ln(1 + r^2). \tag{27}$$

*and observes the following property*

$$\uparrow r_i \implies \uparrow S(r_i) = D_{KL}(p_{z_i}(v) \| \mathcal{N}(v; 0, 1)). \tag{28}$$

*Proof.* We estimate the Kullback–Leibler divergence between $p_{z_i}(v)$ and $\mathcal{N}(v; 0, 1)$:

$$D_{KL}(p_{z_i}(v) \| \mathcal{N}(v; 0, 1)) = \int_{-\infty}^{\infty} p_{z_i}(v) \ln \frac{p_{z_i}(v)}{\mathcal{N}(v; 0, 1)} dv$$

$$= -H(p_{z_i}(v)) + H((p_{z_i}(v), \mathcal{N}(v; 0, 1)))$$

$$= -H(p_{z_i}(v)) + \frac{1}{2}(1 + \ln 2\pi), \tag{29}$$

As the funcion $\mathcal{N}(v;0,1)))$ is given by

$$\mathcal{N}(v;0,1))) = \frac{1}{\sqrt{2\pi}} e^{-\frac{v^2}{2}}, \tag{30}$$

then the term $H((p_{z_i}(v), \mathcal{N}(v;0,1)))$ can be formulated as

$$
\begin{aligned}
H((p_{z_i}(v), \mathcal{N}(v;0,1))) &= \int_{-\infty}^{\infty} p_{z_i}(v) \ln \frac{1}{\mathcal{N}(v;0,1)} dv \\
&= \int_{-\infty}^{\infty} p_{z_i}(v) \Big[ \frac{1}{2}\ln 2\pi + \frac{1}{2}v^2 \Big] dv \\
&= \frac{1}{2}\ln 2\pi \int_{-\infty}^{\infty} p_{z_i}(v) dv + \frac{1}{2}\int_{-\infty}^{\infty} v^2 p_{z_i}(v) dv \\
&= \frac{1}{2}\ln 2\pi + \frac{1}{2}\int_{-\infty}^{\infty} v^2 \frac{p_0(v)+p_1(v)}{2} dv \\
&= \frac{1}{2}\ln 2\pi + \frac{1}{4}\Big[ E_{v\sim p_0(v)}[v^2] + E_{v\sim p_1(v)}[v^2] \Big] \\
&= \frac{1}{2}\ln 2\pi + \frac{1}{4}\Big[ E_{p_0(v)}[v]^2 + Var_{p_0(v)}[v] + E_{p_1(v)}[v]^2 + Var_{p_1(v)}[v] \Big] \\
&= \frac{1}{2}\ln 2\pi + \frac{1}{2}
\end{aligned}
\tag{31}
$$

As the entropy $H(p)$ is concave in the probability mass function $p$, a lower bound of $H(p_{z_i})$ is given by:

$$
\begin{aligned}
H(p_{z_i}(v)) &= H(\frac{p_0(v)+p_1(v)}{2}) \\
&\geq \frac{1}{2}H(p_0(v)) + \frac{1}{2}H(p_1(v)) \\
&= \frac{1}{2}H(\mathcal{N}(v; -\frac{r}{\sqrt{1+r^2}}, \frac{1}{\sqrt{1+r^2}})) + \frac{1}{2}H(\mathcal{N}(v; \frac{r}{\sqrt{1+r^2}}, \frac{1}{\sqrt{1+r^2}})) \\
&= \frac{1}{2}\Big[ \frac{1}{2}(1+\ln(2\pi(\frac{1}{\sqrt{1+r^2}})^2)) + \frac{1}{2}(1+\ln(2\pi(\frac{1}{\sqrt{1+r^2}})^2)) \Big] \\
&= \frac{1}{2}(1+\ln 2\pi) - \frac{1}{2}\ln(1+r^2).
\end{aligned}
\tag{32}
$$

The upper bound of $H(p_{z_i})$ is given by:

$$
\begin{aligned}
H(p_{z_i}) &= -\int_{-\infty}^{\infty} p_{z_i}(v) \ln p_{z_i}(v) dv \\
&= -\int_{-\infty}^{\infty} \frac{p_o(v)+p_1(v)}{2} \ln \frac{p_o(v)+p_1(v)}{2} dv \\
&= -\frac{1}{2}\Big[ \int_{-\infty}^{\infty} p_o(v)[\ln \frac{p_o(v)}{2} + \ln(1+\frac{p_1(v)}{p_0(v)})] dv + \int_{-\infty}^{\infty} p_1(v)[\ln \frac{p_1(v)}{2} + \ln(1+\frac{p_0(v)}{p_1(v)})] dv \Big] \\
&\leq -\frac{1}{2}\Big[ \int_{-\infty}^{\infty} p_o(v) \ln \frac{p_o(v)}{2} dv + \int_{-\infty}^{\infty} p_1(v) \ln \frac{p_1(v)}{2} dv \Big] \\
&= \frac{1}{2}\Big[ H(p_0) + H(p_1) + 2\ln 2 \Big] \\
&= \frac{1}{2}(1+\ln 2\pi) - \frac{1}{2}\ln(1+r^2) + \ln 2.
\end{aligned}
\tag{33}
$$

Thus, the Kullback–Leibler divergence is bounded by :

$$\frac{1}{2}\ln(1+r^2) - \ln 2 \leq D_{KL}(p_{z_i}(v)\|\mathcal{N}(v;0,1)) \leq \frac{1}{2}\ln(1+r^2). \tag{34}$$

Since the lower and upper bounds differ by a constant term, and the lower bound increases significantly as $r$ rises, the Kullback–Leibler divergence is asymptotically tightly bounded by:

$$D_{KL}(p_{z_i}(v)\|\mathcal{N}(v;0,1)) = \Theta(\ln(1 + r^2)) \tag{35}$$

### A.4 PROOF OF PROPOSITION 2

**Proposition 2 (restated)** *The conditional entropy of a Gaussian mixture $\boldsymbol{v}_s$ of $\mathcal{G}_s$ in Eq. 19 is given by*

$$H(\boldsymbol{v}_s|y_i) = \frac{D}{2}(1 + \ln(2\pi)) + \frac{1}{2}\ln|\boldsymbol{\Sigma}_s|, \tag{36}$$

*where $D$ is the dimensions of the features. If each feature $v_s^d$ is independent, then*

$$H(\boldsymbol{v}_s|y_i) = \frac{D}{2}(1 + \ln(2\pi)) + \sum_{d=1}^{D}\ln\sigma_s^d. \tag{37}$$

*Proof.* For the variable follows a Gaussian distribution:

$$\boldsymbol{v} \sim \mathcal{N}_D(\boldsymbol{\mu}, \boldsymbol{\Sigma}), \tag{38}$$

The derivation of its entropy is given by

$$
\begin{aligned}
H(\boldsymbol{v}) &= -\int p(\boldsymbol{v})\ln p(\boldsymbol{v})d\boldsymbol{v} \\
&= -\mathbb{E}\big[\ln\mathcal{N}_D(\boldsymbol{\mu}, \boldsymbol{\Sigma})\big] \\
&= -\mathbb{E}\Big[\ln\big[(2\pi)^{-\frac{D}{2}}|\boldsymbol{\Sigma}|^{-\frac{1}{2}}\exp(-\frac{1}{2}(\boldsymbol{v}-\boldsymbol{\mu})^{\top}\boldsymbol{\Sigma}^{-1}(\boldsymbol{v}-\boldsymbol{\mu}))\big]\Big] \\
&= \frac{D}{2}\ln(2\pi) + \frac{1}{2}\ln|\boldsymbol{\Sigma}| + \frac{1}{2}\mathbb{E}\big[(\boldsymbol{v}-\boldsymbol{\mu})^{\top}\boldsymbol{\Sigma}^{-1}(\boldsymbol{v}-\boldsymbol{\mu}))\big] \\
&\overset{*}{=} \frac{D}{2}(1 + \ln(2\pi)) + \frac{1}{2}\ln|\boldsymbol{\Sigma}|
\end{aligned}
\tag{39}
$$

Step $*$ is a little trickier. It relies on several properties of the trace operator:

$$
\begin{aligned}
\mathbb{E}\big[(\boldsymbol{v}-\boldsymbol{\mu})^{\top}\boldsymbol{\Sigma}^{-1}(\boldsymbol{v}-\boldsymbol{\mu}))\big] &= \mathbb{E}\Big[\mathrm{tr}\big[(\boldsymbol{v}-\boldsymbol{\mu})^{\top}\boldsymbol{\Sigma}^{-1}(\boldsymbol{v}-\boldsymbol{\mu}))\big]\Big] \\
&= \mathbb{E}\Big[\mathrm{tr}\big[\boldsymbol{\Sigma}^{-1}(\boldsymbol{v}-\boldsymbol{\mu})(\boldsymbol{v}-\boldsymbol{\mu})^{\top}\big]\Big] \\
&= \mathrm{tr}\Big[\boldsymbol{\Sigma}^{-1}\mathbb{E}\big[(\boldsymbol{v}-\boldsymbol{\mu})(\boldsymbol{v}-\boldsymbol{\mu})^{\top}\big]\Big] \\
&= \mathrm{tr}\Big[\boldsymbol{\Sigma}^{-1}\boldsymbol{\Sigma}\Big] \\
&= \mathrm{tr}(\boldsymbol{I}_D) \\
&= \frac{D}{2}
\end{aligned}
\tag{40}
$$

## B DETAILED IMPLEMENTATION

### B.1 KLD LOSS

For the implementation of KLD loss in Eq. 3 and Eq. 14, we follows the widely-used version from Kingma & Welling (2014). The detailed loss formulation is given

$$\mathrm{KLD}(\boldsymbol{z}, \mathcal{N}(0, \boldsymbol{I})) = -\frac{1}{2}\sum_{j=1}^{J}(1 + \log(\sigma_j)^2 - (\mu_j)^2 - (\sigma_j)^2), \tag{41}$$

where $\boldsymbol{z} = \boldsymbol{\mu} + \boldsymbol{\sigma} \odot \boldsymbol{\epsilon}$, and $\boldsymbol{\epsilon} \sim \mathcal{N}(0, \boldsymbol{I})$.

## B.2   D-VAE

In the implementation of D-VAE, the encoder comprises 7 convolutional layers with Batch Normalization, while the decoder for both branches consists of 4 convolutional layers with Instance Normalization. To predict the mean $\mu$ and standard deviation $\sigma$, we employ one convolutional layer with a kernel size of 1 for each variable.

During the training of D-VAE, we configure the number of training epochs to be 60 for CIFAR-10 and CIFAR-100. However, for ImageNet, which involves significant computational demands, we limit the training epochs to 20. It's important to note that we do not use any transformations on the training data when training D-VAEs. For D-VAE training on poisoned CIFAR-10/100, we use a KLD limit of 1.0 in the first stage and 3.0 in the second stage, with only a single $\times 0.5$ downsampling to preserve image quality. For ImageNet, which has higher-resolution images, we employ more substantial downsampling ($\times 0.125$) in the first stage and set a KLD limit of 1.5, while the second stage remains the same as with CIFAR. When comparing the poisoned input and the reconstructed output, these hyperparameters yield PSNRs of around 28 for CIFAR and 30 for ImageNet.

## C   DETAILED IMPLEMENTATION OF THE ATTACK METHODS AND COMPETING DEFENSES

As previous papers may have used varying code to generate perturbations and implemented defenses based on different codebases, we have re-implemented the majority of the attack and defensive methods by referencing their original code resources. In cases where the original paper did not provide code, we will specify the sources we used for implementation.

### C.1   PERTURBATIVE AVAILABILITY POISONING ATTACK METHODS

**NTGA.** For the implementation of NTGA poisoning attacks, we directly download the read-to-use poisoned dataset from the official source of NTGA (Yuan & Wu, 2021).

**EM, TAP, and REM.** For the implementation of EM (Huang et al., 2021), TAP (Fowl et al., 2021), and REM (Fu et al., 2022) poisoning attacks, we follow the official code of REM (Fu et al., 2022).

**SEP.** For the implementation of SEP (Chen et al., 2023) poisoning attacks, we follow the official code of SEP (Chen et al., 2023).

**LSP.** For the implementation of LSP (Yu et al., 2022) poisoning attacks, we follow the official code of LSP (Yu et al., 2022). Particularly, we set the patch size of the colorized blocks to 8 for both CIFAR-10, CIFAR-100, ImageNet-subset.

**AR.** For the implementation of AR poisoning attacks, we directly download the read-to-use poisoned dataset from the official source of AR (Sandoval-Segura et al., 2022).

**OPS.** For the implementation of OPS. (Wu et al., 2023) poisoning attacks, we follow the official code of OPS. (Wu et al., 2023).

### C.2   COMPETING DEFENSES

**Image shortcut squeezing (ISS).** For the implementation of ISS (Liu et al., 2023), which consists of bit depth reduction (depth decreased to 2), grayscale (using the official implementation by torchvision.transforms), JPEG compression (quality set to 10), we follow the official code of ISS (Liu et al., 2023). Although most of the reported results align closely with the original paper's findings, we observed that EM and REM poisoning attacks generated using the codebase of REM (Fu et al., 2022) display a notable robustness to Grayscale, which differs somewhat from the results reported in the original paper.The unreported results for the performance of each compression on the CIFAR-100 and ImageNet datasets are presented in Table D.

**Adversarial training (AT).** For the implementation of adversarial training, we follow the official code of pgd-AT (Madry et al., 2018) with the adversarial perturbation subject to $\ell_\infty$ bound, and set $\epsilon = \frac{8}{255}$, iterations $T = 10$, and step size $\alpha = \frac{1.6}{255}$.

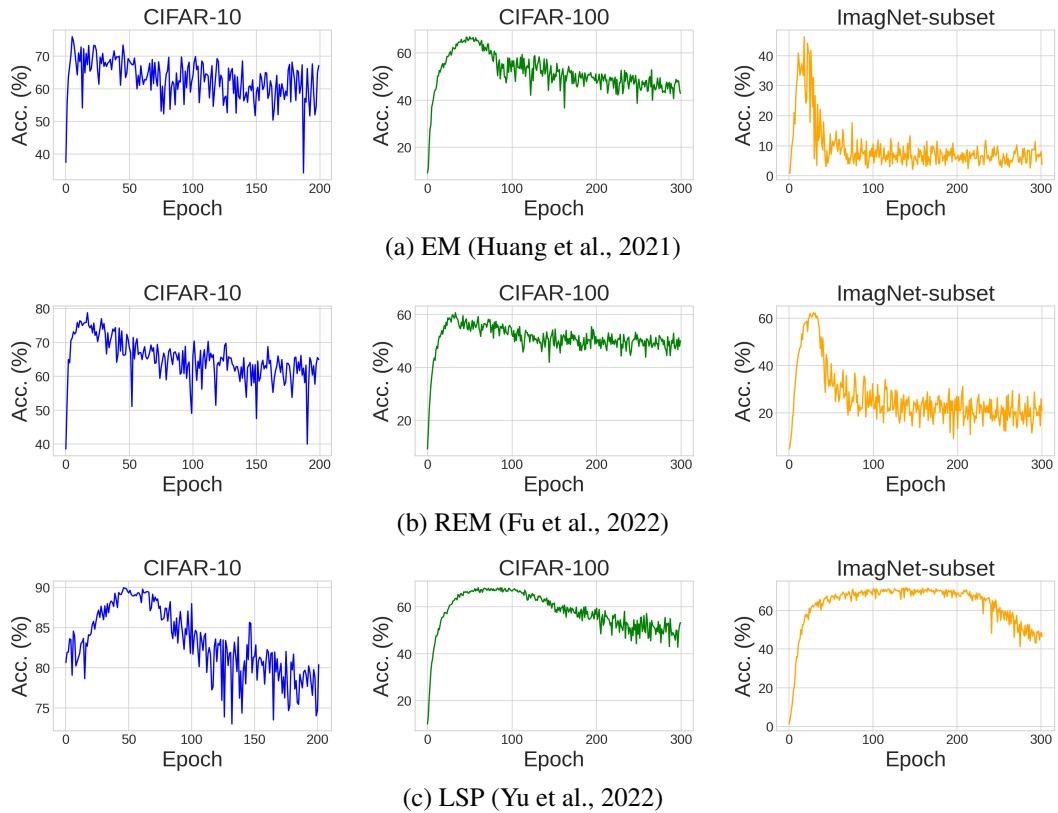

Figure 3: Test accuracy (%) for each training epoch when using adversarial augmentation (Qin et al., 2023b)

**AVATAR.** In our implementation of AVATAR, which employs a diffusion model trained on the clean CIFAR-10 dataset to purify poisoned samples, we utilized the codebase from a benchmarking paper (Qin et al., 2023a). This choice was made since AVATAR (Dolatabadi et al., 2023) does not offer official implementations.

**Adversarial augmentations (AA).** In our implementation of AA, we utilized the codebase from the original paper (Qin et al., 2023b). AA comprises two stages. In the first stage, loss-maximizing augmentations are employed for training, with a default number of repeated samples set to K = 5. In the second stage, a lighter augmentation process is applied, with K = 1. In all experiments conducted on CIFAR-10, CIFAR-100, and the 100-class ImageNet subset, we strictly adhere to the same hyperparameters as detailed in the original paper. Nevertheless, we observed that this training-time method can partially restore the test accuracy if we report the highest accuracy achieved among all training epochs. However, it's worth noting that the model may still exhibit a tendency to overfit to the shortcut provided by the poisoned samples. Consequently, this can lead to a substantial drop in test accuracy during the second stage, which employs lighter augmentations. The test accuracy for each training epoch is depicted in Figure 3. Additionally, we have included the best accuracy for AA in Table D. It's notable that our results from the last epoch surpass the performance of AA, showcasing the superiority.

## D  VISUAL RESULTS

In this section, we present visual results of the purification process on the ImageNet-subset. As depicted in Figure 4, the purification carried out during stage 1 is effective in removing a significant portion of poison patterns, particularly for LSP poisoning attacks. The remaining poisoning attacks are subsequently eliminated in stage 2, resulting in completely poison-free data.

Table 9: Test acc. (%) of models trained on CIFAR-10 poisoning attacks.

| Norm | Attacks | w/o | AA | Ours |
|---|---|---|---|---|
| | Clean | **94.57** | 92.66 | 93.29 |
| $\ell_\infty = 8$ | NTGA | 11.10 | 86.35 | **89.21** |
| | EM | 12.26 | 76.00 | **91.42** |
| | TAP | 25.44 | 71.56 | **90.48** |
| | REM | 22.43 | 78.77 | **86.38** |
| | SEP | 6.63 | 71.95 | **90.74** |
| $\ell_2 = 1.0$ | LSP | 13.14 | 89.97 | **91.20** |
| | AR | 12.50 | 67.61 | **91.77** |
| $\ell_0 = 1$ | OPS | 22.03 | 72.54 | **88.95** |

Table 10: Test acc. (%) of models trained on CIFAR-100 poisoning attacks.

| Attacks | w/o | AA | BDR | Gray | JPEG | Ours |
|---|---|---|---|---|---|---|
| Clean | **77.61** | 70.22 | 63.52 | 71.59 | 57.85 | 70.72 |
| EM | 12.30 | 66.84 | 61.91 | 48.83 | 58.08 | **68.79** |
| TAP | 13.44 | 49.36 | 55.09 | 9.69 | 57.33 | **65.54** |
| REM | 16.80 | 60.74 | 57.51 | 55.99 | 58.13 | **68.52** |
| SEP | 4.66 | 37.73 | 31.95 | 4.47 | 57.76 | **64.02** |
| LSP | 2.91 | 68.22 | 22.13 | 44.18 | 53.06 | **67.73** |
| AR | 2.71 | 44.32 | 29.68 | 23.09 | 56.60 | **63.73** |
| OPS | 12.56 | 40.20 | 11.56 | 19.33 | 54.45 | **65.10** |

Table 11: Test acc. (%) of models trained on ImageNet subset poisoning attacks.

| Attacks | w/o | AA | BDR | Gray | JPEG | Ours |
|---|---|---|---|---|---|---|
| Clean | **80.52** | 73.66 | 75.84 | 76.92 | 72.90 | 76.78 |
| EM | 1.08 | 46.30 | 2.78 | 14.02 | 72.44 | **74.80** |
| TAP | 12.56 | 72.10 | 45.74 | 33.66 | 73.24 | **76.56** |
| REM | 2.54 | 62.30 | 57.51 | 55.99 | 58.13 | **72.56** |
| LSP | 2.50 | 71.72 | 22.13 | 44.18 | 53.06 | **76.06** |

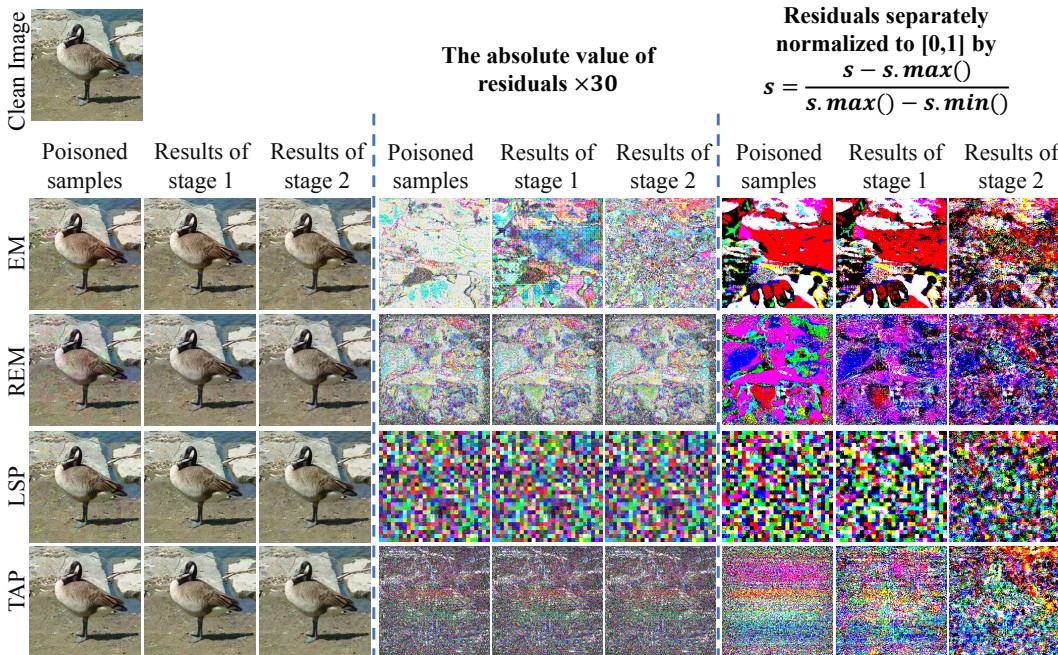

Figure 4: Visual results of images before/after purification. Results of stage 2 denote the final purified results. The image is from ImageNet-subset, and the residuals to the clean images are normalized by two ways.

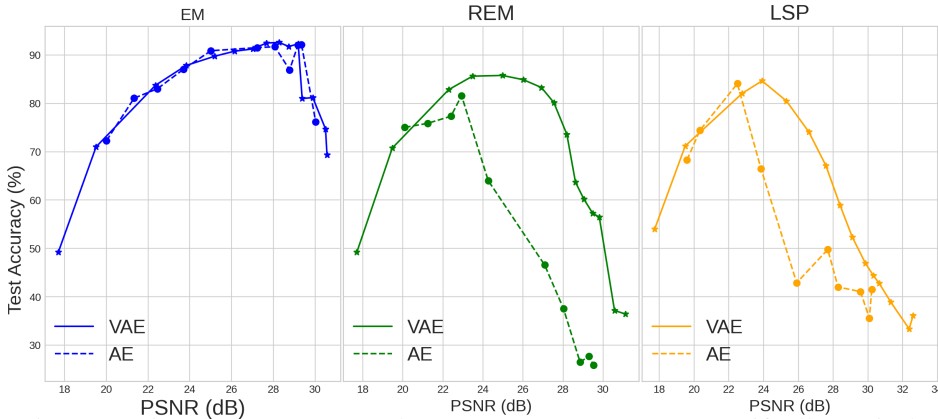

Figure 5: Comparison between VAEs and AEs: PSNR Vs. Test Acc. Specifically, we include EM, REM, and LSP as attack methods here.

# E    COMPARISON WITH NON-VARIATIONAL AUTO-ENCODERS

In this section, we conduct experiments on purification using non-variational auto-encoders (AEs) with an information bottleneck. To achieve non-variational auto-encoders with different bottleneck levels, we modify the width of the features within the auto-encoder architecture. This results in models with varying parameter numbers. Then, we proceed to train the AE on the poisoned CIFAR-10 dataset, and test on the clean test dataset with classifiers trained on the purified dataset. As depicted in Figure 5, when considering the similar level of reconstruction quality measured by PSNR, VAEs exhibit a greater capacity to remove poison patterns in both the REM and LSP poisoning attacks. However, for EM poisoning attacks, the outcomes are comparable. These observations align with the theoretical analysis presented in Section 3.3.

Table 12: Computation requirement of the proposed methods.

| Component | Train D-VAE for twice | Perform inference on the poisoned data three times | Train a classifier | Total Time |
|---|---|---|---|---|
| Our method | 23 minutes | less than 2 minutes | 16 minutes | 41 minutes |
| Adversarial Training | N.A. | N.A. | 229 minutes | 229 minutes |

Table 13: Results using JPEG with various quality settings. The experiments are on CIFAR-10 dataset.

| Defenses /Attacks | JPEG (quality 10) PSNR 22 | JPEG (quality 30) PSNR 25 | JPEG (quality 50) PSNR 27 | JPEG (quality 70) PSNR 28 | Ours PSNR 28 |
|---|---|---|---|---|---|
| NTGA | 78.97 | 66.83 | 64.28 | 60.19 | **89.21** |
| EM | 85.61 | 70.48 | 54.22 | 42.23 | **91.42** |
| TAP | 84.99 | 84.82 | 77.98 | 57.45 | **90.48** |
| REM | 84.40 | 77.73 | 71.19 | 63.39 | **86.38** |
| SEP | 84.97 | 87.57 | 82.25 | 59.09 | **90.74** |
| LSP | 79.91 | 42.11 | 33.99 | 29.19 | **91.20** |
| AR | 84.97 | 89.17 | 86.11 | 80.01 | **91.77** |
| OPS | 77.33 | 79.01 | 68.68 | 59.81 | **88.96** |
| Mean | 78.89 | 74.71 | 67.33 | 56.42 | **90.02** |

# F    COMPUTATION AND COMPARISON WITH JPEG COMPRESSION

In this section, we present the computation requirement and the compassion with JPEG compression. The Table 12 below presents the training time for D-VAE, the inference time for the poisoned dataset, and the time to train a classifier using the purified dataset. For comparison, we include the training-time defense **Adversarial Training**. It's important to note that the times are recorded using CIFAR-10 as the dataset, PyTorch as the platform, and a single Nvidia RTX 3090 as the GPU. As can see from the results, the total purification time is approximately one and a half times longer than training a classifier, which is acceptable. Compared to adversarial training, our methods are about 5 times faster. Additionally, our method achieves an average performance around 90%, which is 15% higher than the performance achieved by adversarial training.

We also note a limitation in the JPEG compression approach used in ISS (Liu et al., 2023)—specifically, they set the JPEG quality to 10 to purify poisoned samples, resulting in significant image degradation. In the Table 13, we present results using JPEG with various quality settings. Notably, our proposed methods consistently outperform JPEG compression when applied at a similar level of image corruption. Therefore, in the presence of larger perturbation bounds, JPEG may exhibit sub-optimal performance. Moreover, our method excels in eliminating the majority of poison patterns in the first stage, rendering it more robust to larger perturbation bounds. Table 5 5 of the main paper illustrates that when confronted with LSP attacks with larger bounds, our method demonstrates significantly smaller performance degradation compared to JPEG (with quality 10), *e.g.,* 86.13 Vs. 41.41 in terms of test accuracy.

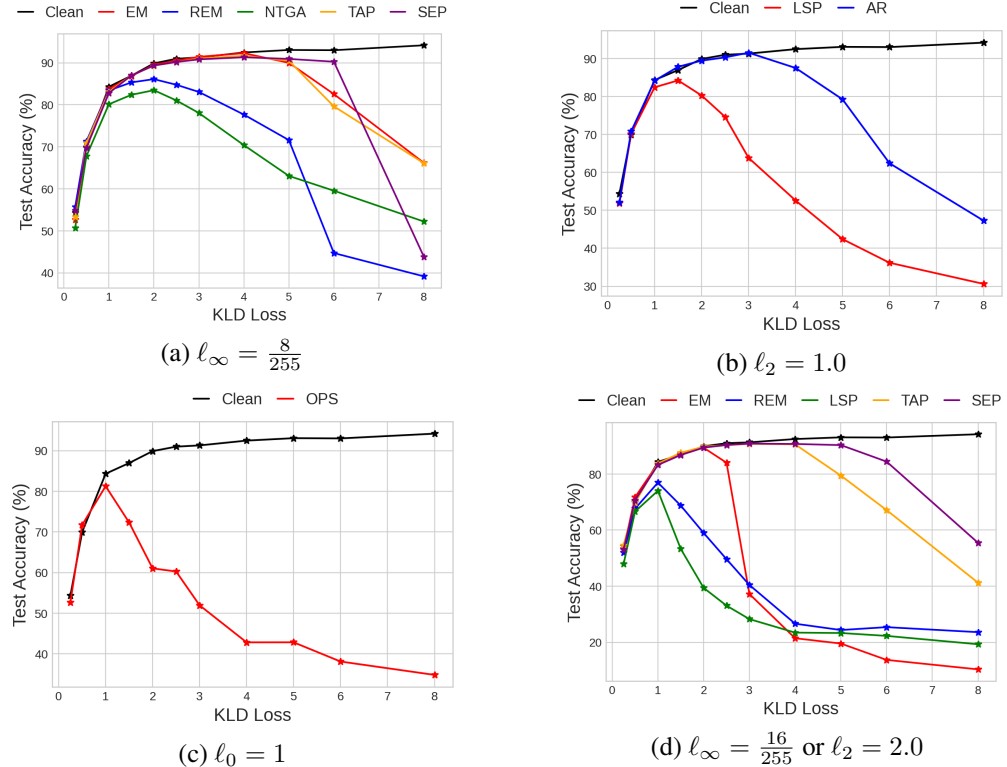

Figure 6: Results using D-VAEs: Test Acc. Vs. KLD Loss is assessed on the poisoned CIFAR-10.

## G    MORE EXPERIMENTS ON TRAINING D-VAE ON ATTACK METHODS WITH VARIOUS UPPER BOUNDS ON THE KLD LOSS

Some concerns regarding whether a sizable component of the perturbation will end up being learned into $\hat{x}$ may arise in certain cases, such as when the upper bound on the KLD loss is not set low. Nevertheless, when the KLD loss is set to a low value, the presence of poison patterns in the reconstructed $\hat{x}$ is shown to be minimal. This observation is supported by both empirical experiments in Section 3.2 and theoretical explanations provided in Section 3.3. These outcomes are primarily attributed to the fact that the reconstruction of $\hat{x}$ depends on the information encoded in the latent representation $z$, *i.e.,* $\hat{x}$ is directly generated from $z$ using a decoder. The theoretical insights discussed in Section 3.3 highlight that **Theorem 1** indicates that perturbations which create strong attacks tend to have a larger inter-class distance and a smaller intra-class variance. Additionally, **Theorem 2** and **Remark 1** indicate that poison patterns possessing these characteristics are more likely to be eliminated when aligning the features with a normal Gaussian distribution (as done by the VAE).

To further validate these observations, we now include additional experiments in Appendix G by training D-VAE on all attack methods with various upper bounds for the KLD loss. Additionally, we have performed experiments on attacks with larger perturbations. Notably, we have added results on the clean dataset for comparison. As depicted in Figure 6, when the upper bound on the KLD loss is set below 1.0, the curves of the results on the poisoned dataset align closely with the results on the clean dataset. Furthermore, as the upper bound decreases, the removal of poison patterns in the reconstructed $\hat{x}$ increases. While it is evident that larger perturbations may be better retained in $\hat{x}$, it is a cat-and-mouse game between defense and attack. Additionally, larger perturbations tend to be more noticeable. These findings affirm that the observations hold for all existing attack methods, and setting a low upper bound (*e.g.,* 1.0, as in the main experiments) on the KLD loss significantly ensures that $\hat{x}$ contains few poison patterns.

