# OpenReview forum: "Purify Perturbative Availability Poisons via Rate-Constrained Variational Autoencoders"
_ICLR.cc/2024/Conference — Submitted to ICLR 2024_

### Official Review · Reviewer_h4wy · 2023-10-28

**Soundness:** 2 fair
**Presentation:** 2 fair
**Contribution:** 2 fair
**Rating:** 5
**Confidence:** 2

**Summary:**

The authors uncover and theoretically explain that D-VAE can effectively purify poison patterns. Based on this insight, the authors propose a two-stage purification approach with learnable class-wise embeddings.

**Strengths:**

The authors show that perturbations tend to have a larger inter-class distance and a smaller intra-class variance.

The pre-training processing shows the potential for effective and efficient defence against poison perturbations.

**Weaknesses:**

Concern regarding pseudo-robustness. If we design an adaptive poison pattern, that uses a white-box attack (e.g. PGD) to end-to-end (treat VAE and classifier as a whole) generate perturbations, can this method also be used for effective purification?

Concern regarding generalization capacity. If we use a new type of poisoning attack that has not been seen in class-wise embedding (out of the training poisoning attacks), can this method still effectively purify it?

Concern regarding training and inference costs. Does the training set of this method include all poisoning attacks? What is the cost of training? Three additional forward propagations are required in the algorithm, what is the inference time?

**Questions:**

Please refer to the questions in the Weaknesses.

---

> ### Author Response · Authors · 2023-11-16
> **Official Comments by Authors**
>
> Thank you for the valuable suggestions! I have submitted a revised draft addressing the comments provided by the reviewers. Below, you'll find the responses to the questions:
>
> **Q1.** Concern regarding pseudo-robustness. If we design an adaptive poison pattern, that uses a white-box attack (e.g. PGD) to end-to-end (treat VAE and classifier as a whole) generate perturbations, can this method also be used for effective purification?
>
> **A1.** This paper primarily addresses the defense against perturbative availability poisoning attacks, a type of data poisoning attack. Typically, such attacks occur during the data collection process or involve malicious modifications to the dataset before model training. Consequently, as the attacker has no access to model training, defense strategies against data poisoning attacks assume the trustworthiness of the defense process or model training process. As a result, it is generally not feasible for attackers to compromise the VAE during the purification process.
>
>
>
> **Q2.** Concern regarding generalization capacity. If we use a new type of poisoning attack that has not been seen in class-wise embedding (out of the training poisoning attacks), can this method still effectively purify it?
>
> **A2.** In research focusing on defenses against data poisoning attacks, the defense process is trustworthy, and attackers have no access after poisoning the training dataset. Once we obtains the poisoned dataset, we can proceed to train a D-VAE. Then, at inference time, the dataset to be purified has already been seen by D-VAE during training, eliminating the possibility of encountering a new type of poisoning attack that D-VAE has not been exposed to. The reviewer's suggested settings are indeed intriguing. However, given the constraints of the rebuttal timeline, we will consider and discuss these aspects in our future work.
>
>
>
> **Q3.** Concern regarding training and inference costs. Does the training set of this method include all poisoning attacks? What is the cost of training? Three additional forward propagations are required in the algorithm, what is the inference time?
>
> **A3.** The training dataset for our methods is the poisoned dataset, constructed using a single poisoning attack method. Consequently, for each poisoning attack, we train a distinct D-VAE for purification.
>
> The table below presents the training time for D-VAE, the inference time for the poisoned dataset, and the time to train a classifier using the purified dataset. For comparison, we include the training-time defense **Adversarial Training**. It's important to note that the times are recorded using CIFAR-10 as the dataset, PyTorch as the platform, and a single Nvidia RTX 3090 as the GPU. As can see from the results, the total purification time is approximately one and a half times longer than training a classifier. Compared to adversarial training, our methods are about 5 times faster. Additionally, our method achieves an average performance around 90%, which is 15% higher than the performance achieved by adversarial training.
>
> |      Component       | Train D-VAE for twice | Perform inference on the poisoned data three times | Train a classifier | Total Time  |
> | :------------------: | :-------------------: | :------------------------------------------------: | :----------------: | :---------: |
> |      Our method      |      23 minutes       |                less than 2 minutes                 |     16 minutes     | 41 minutes  |
> | Adversarial Training |         N.A.          |                        N.A.                        |    229 minutes     | 229 minutes |

---

> ### Author Response · Authors · 2023-11-21
>
> Dear Reviewer h4wy,
>
> Thank you for taking the time to review our submission and providing us with constructive comments and a favorable recommendation. We would like to know if our responses adequately addressed your earlier concerns. Additionally, if you have any further concerns or suggestions, we would be more than happy to address and discuss them to enhance the quality of the paper. We eagerly await your response and look forward to hearing from you.
>
> Best regards,
>
> The authors

---

> > ### Comment · Reviewer_h4wy · 2023-11-23
> >
> > The author seems unable to effectively solve my concerns, this discrepancy may stem from different research topics, so I am willing to reduce the confidence to 2.

---

### Official Review · Reviewer_PyJm · 2023-10-29

**Soundness:** 3 good
**Presentation:** 3 good
**Contribution:** 3 good
**Rating:** 6
**Confidence:** 4

**Summary:**

This paper proposes a defense mechanism against unlearnable examples, a type of attack that aims to decrease the overall utility of a dataset by performing perturbation on the entire dataset. The authors discover that VAEs are capable of removing the perturbations and provide theoretical analysis for justification. The experimental results show SOTA results compared with other baseline defenses against several unlearnable example methods.

**Strengths:**

Defense against unlearnable examples is a difficult problem, e.g., existing data sanitization techniques cannot be applied as the entire dataset is perturbed. This paper proposes a novel solution for data purification by using generative models (in particular, VAE) to disentangle the perturbation and clean data. The approach seems promising both theoretically and empirically.

**Weaknesses:**

When studying unlearnable examples, it seems that nowadays people are using three terms: "unlearnable examples", "availability attacks" and "indiscriminate attacks" interchangeably. I am not totally against it, but I think the authors need to be extra careful when specifying the threat model they are considering.

To illustrate more, availability attack is a generally broader definition. For example, in Biggio et al., 2012 and the following works, availability attacks refer to data poisoning attacks that aim to decrease the model performance overall. Specifically, an attacker usually injects poisoned points into the clean training set to achieve such a goal. Later, Huang et al., 2021 propose another type of availability attack that perturbs the entire training set, which is a data protection technique, also known as unlearnable examples. Obviously, these two types of attacks are very different in terms of the attack budget and attack purpose. I guess "perturbation availability poison" (in the title) seems to be a good definition, but I suggest the authors use a similar explicit definition throughout the paper.

In summary, I encourage the authors to revisit the literature on availability attacks, explicitly identify the threat model they are considering, and specify it in Section 2 while respectively discussing other works (e.g., other data poisoning attacks: targeted attacks, backdoor attacks, and especially indiscriminate attacks). Specifically, sentences like the following one need to be rewritten to reflect the true progress of this field:
> In the realm of DNNs, the majority of existing research
has primarily concentrated on backdoor attacks. However, there has been a growing interest in
availability attacks, also named unlearnable examples, prompting the exploration to prevent unauthorized model training.

Here, the claim is not supported by any reference and is simply not true: there are numerous other poisoning attacks considering DNNs. Also, the second sentence is not accurate, see my arguments above.

**Questions:**

(1) The auxiliary decoder is not very clear to me, and I expect the authors to illustrate more regarding the following aspects:
- What is $u_c$? The authors mention that this is a trainable class-wise embedding, but it is not clear how it is initialized, how it is trained, and why it helps the decoder $D_{\theta_p}$ to generate the disentangled perturbation.
- How do you optimize $\theta_p$? Normally, to train a VAE model, one aims to reconstruct images, in this case, the perturbation $p$. However, to my understanding, a defender should not have access to the perturbations, then how do you construct the loss for $\theta_p$?

---

> ### Author Response · Authors · 2023-11-16
> **Official Comments by Authors (Part 1)**
>
> Thank you for the valuable suggestions! I have submitted a revised draft addressing the comments provided by the reviewers. Below, you'll find the responses to the questions:
>
> **Q1.** Offer a more precise definition of the attacks and a more comprehensive discussion, including detailed information on the availability attacks in Section 2.
>
> **A1.** We have extensively revised the **Data Poisoning** in Section 2, presenting a thorough discussion of other types of attacks related to perturbative availability poisoning attacks. Perturbative availability poisoning attacks aim to degrade the overall performance on validation and test dataset. The attacker has the capability to poison the entire training data using bounded perturbations, *e.g.,* $\Vert \pmb{p} \Vert_{\infty} \le \frac{8}{255}$, while keeping the labels unchanged.

---

> > ### Comment · Reviewer_PyJm · 2023-11-16
> >
> > Thank you for the rebuttal and revision. The revised Section 2 looks better to me, however, there still exist some problems that should be easy to address:
> >
> > (1) The authors mention that:
> > > Typically, poisoned samples are notably modified or replaced with newly
> > generated samples, and take only a fraction of the entire dataset.
> >
> > This statement is not accurate namely that for such attacks, the attacker cannot "modify" or "replace" any clean training data as the assumption would be too strong (of course, there exists replacing attacks in the literature, but is not considered here). The author should be aware that this specific threat model only considers **adding** poisoned data on top of clean data.
> >
> > (2) Next the authors write:
> > > However, these methods often fail to sufficiently degrade model performance
> > (e.g., only a 16% drop on CIFAR-10), and the poisoned samples are easily distinguishable.
> >
> > These statements are not entirely wrong but I don't think they are good motivations to study unlearnable examples as the attack budgets are very different. To illustrate more, e.g., Koh & Liang consider adding only 3% of poisoned data while UE considers modifying 100% of the clean set. Thus considering the much lower poisoning fraction, a 16% accuracy drop is probably not tiny.
> >
> > As a result, as it is not a fair comparison, I would suggest the authors introduce these two methods in parallel as separate categories, one as an attack method, and one as a protection technique.

---

> > > ### Author Response · Authors · 2023-11-16
> > > **Official Comments by Authors**
> > >
> > > Thank you for the feedback! I've submitted a revised draft with an updated Section 2.

---

> > > > ### Comment · Reviewer_PyJm · 2023-11-16
> > > >
> > > > Thank you for the rapid reply. I am very satisfied with the additional round of revision and I have raised my score accordingly.

---

> ### Author Response · Authors · 2023-11-16
> **Official Comments by Authors (Part 2)**
>
> **Q2.** The role of the auxiliary decoder and the process of initializing and optimizing $\pmb{u_y}$, as well as optimizing $\pmb{u_y}$ to assist $D_{\theta_p}$ in generating the disentangled perturbation $\pmb{\hat{p}}$, are not explicitly explained. Additionally, the optimization of $\theta_p$ in the absence of groundtruth values for the added perturbations $\pmb{p}$ is unclear.
>
> **A2.** We have revised and provided a detailed explanation of the components in Section 3.4, focusing on the learning process of disentangled perturbations.
>
> - Initialization of $\pmb{u_y}$.
>   - Given that the size of the input $\pmb{x}$ for each dataset is fixed, such as $3 \times 32 \times 32$ for the CIFAR-10 dataset, the size of the latents $\pmb{z}$ is fixed at $128 \times 16 \times 16$. Consequently, $\pmb{u_y}$ for each class is of the same size as $\pmb{z}$ and is defined and randomly initialized using `torch.nn.Embedding` when utilizing PyTorch as the platform.
>
> - Optimize $\pmb{u_y}$ and $\theta_p$ in the absence of groundtruth values for the added perturbations $\pmb{p}$.
>
>   - Given that the defender lacks groundtruth values for the perturbations $\pmb{p}$, it is not possible to optimize $\pmb{u_y}$ and $D_{\theta_p}$ to learn to predict $\pmb{\hat{p}}$ directly by minimizing ${\lVert \pmb{p} - \pmb{\hat{p}} \rVert}_2^2$ during model training.
>
>   - Expanding on the insights from Section 3.2 and **Remark 1** in Section 3.3, when imposing a low upper bound on the KLD loss, creating an information bottleneck on the latents $\pmb{z}$, the reconstructed $\pmb{\hat{x}}$ cannot achieve perfect reconstruction, making poison patterns more challenging to be recovered in $\pmb{\hat{x}}$. As a result, a significant portion of poison patterns $\pmb{p}$ persists in the residuals $\pmb{x} - \pmb{\hat{x}}$.
>
>   - Following **Proposition 2** in Section 3.3, the majority of poison patterns associated with each class data exhibit relatively low entropy, suggesting that they can be predominantly reconstructed using representations with limited capacity. Considering that most poison patterns are crafted to be sample-wise, we propose a learning approach that maps the summation of a trainable class-wise embedding $\pmb{u_y}$ and the latents $\pmb{z}$ to $\pmb{\hat{p}}$ through an auxiliary decoder $D_{\theta_p}$.
>
>   - To learn $\pmb{u_y}$ and train $D_{\theta_p}$, we propose minimizing ${\lVert (\pmb{x} - \pmb{\hat{x}}) - \pmb{\hat{p}} \rVert}_2^2$, as the residuals $\pmb{x} - \pmb{\hat{x}}$ contain the majority of the groundtruth $\pmb{p}$ when imposing a low upper bound on the KLD loss.
>
> - Next, let's analyze why ${\lVert \pmb{x} - \pmb{\hat{x}} \rVert}_2^2$ and ${\lVert (\pmb{x} - \pmb{\hat{x}}) - \pmb{\hat{p}} \rVert}_2^2$ are not contradictory.
>   - As there is a information bottleneck in the latents $\pmb{z}$ and $\pmb{u_y}$, we assume that the best-reconstructed images are represented by $\pmb{\tilde{x}}$, and the best-reconstructed perturbations are represented by $\pmb{\tilde{p}}$.
>   - If some parts of $\pmb{\tilde{x}}$ are not recovered in $\pmb{\hat{x}}$, we denote $\pmb{\hat{x}} = \pmb{\tilde{x}} - \pmb{\Delta{x}}$.
>
>   - As $\pmb{\hat{p}}$ is generated by $\pmb{u_y}+\pmb{z}$, the $\pmb{\Delta x}$ could be recovered in $\pmb{\hat{p}}$, leading to $\pmb{\hat{p}} = \pmb{\tilde{p}} + \pmb{\Delta{x}}$.
>
>   - We can observe that ${\lVert \pmb{x} - \pmb{\hat{x}} \rVert}_2^2 + {\lVert (\pmb{x} - \pmb{\hat{x}}) - \pmb{\hat{p}} \rVert}_2^2$ is minimized when $\pmb{{\Delta x}}= \pmb{0}$, as ${\lVert \pmb{x} - \pmb{\hat{x}} \rVert}_2^2 + {\lVert (\pmb{x} - \pmb{\hat{x}}) - \pmb{\hat{p}} \rVert}_2^2 = {\lVert \pmb{x} - \pmb{\tilde{x}} + \pmb{\Delta{x}} \rVert}_2^2 + {\lVert (\pmb{x} - \pmb{\tilde{x}}) - \pmb{\tilde{p}} \rVert}_2^2$ .
>
>   - When $\pmb{{\Delta x}}= \pmb{0}$, ${\lVert \pmb{x} - \pmb{\hat{x}} \rVert}_2^2 + {\lVert (\pmb{x} - \pmb{\hat{x}}) - \pmb{\hat{p}} \rVert}_2^2= {\lVert \pmb{x} - \pmb{\tilde{x}} \rVert}_2^2 + {\lVert (\pmb{x} - \pmb{\tilde{x}}) - \pmb{\tilde{p}} \rVert}_2^2$ .
>
>   - In the table below, we present quantitative results indicating that $\pmb{\hat{x}}$ reconstructed by VAE or D-VAE exhibits similar PSNR when the same upper bound for KLD loss is employed. From the PSNR between $\pmb{x}$ and $\pmb{\hat{x}}$+$\pmb{\hat{p}}$, we observe that $\pmb{\tilde{p}}$ can recover some parts of $\pmb{p}$, making $\pmb{\hat{x}} + \pmb{\hat{p}}$ closer to $\pmb{x}$.
>
>     |                    Model/KLD loss                    |  2.0  |  2.5  |  3.0  |  3.5  |  4.0  |
>     | :--------------------------------------------------: | :---: | :---: | :---: | :---: | :---: |
>     |          VAE: PSNR($\pmb{x},\pmb{\hat{x}}$)          | 25.19 | 26.14 | 27.02 | 27.67 | 28.27 |
>     |         D-VAE: PSNR($\pmb{x},\pmb{\hat{x}}$)         | 25.07 | 26.12 | 26.97 | 27.54 | 28.19 |
>     | D-VAE: PSNR($\pmb{x},\pmb{\hat{x}}$+$\pmb{\hat{p}}$) | 25.64 | 26.71 | 27.70 | 28.42 | 29.07 |

---

> > ### Comment · Reviewer_PyJm · 2023-11-16
> >
> > The explanations are satisfactory to me and I don't have additional questions regarding this.
> >
> > I expect the authors to perform another revision on Section 2 and I am happy to adjust my rating accordingly afterwards.

---

### Official Review · Reviewer_D3kf · 2023-10-30

**Soundness:** 2 fair
**Presentation:** 1 poor
**Contribution:** 2 fair
**Rating:** 5
**Confidence:** 3

**Summary:**

This paper deals with availability poisoning attacks: adversarially modified data is laid as a trap; when ingested into a training pipeline, they negatively impact the model’s performance. The authors propose D-VAE, a variational autoencoder method that disentangles the adversarial perturbation from the underlying original data with a one encoder, two decoder architecture; once disentangled, a model can be trained normally on the reconstructed base image. Experiments are conducted on CIFAR-10, CIFAR-100, and a subset of ImageNet, and compare against training-time defenses and pre-training techniques, against a number of poisoning attacks. Results show that the disentangled perturbations are indeed poison patterns, and that training on the reconstructed base images is less degraded than the baselines.

**Strengths:**

## S1. Disentangled Poison Patterns
One thing I was curious about was how faithfully the disentangled poison pattern is decomposed from the input. From Figure 1, it looked like it was vaguely doing the right thing, but was better for some (OPS, LSP) than others (NTGA). As such, Section 4.2 was helping in showing that the disentangled poison patterns are in fact quite effective. However, I still think it’d be interesting to quantify how far off the disentangled perturbations are from what the actual poison pattern is (or at least visualize the residuals); this would help answer whether the model is actually doing disentanglement, or itself somehow learning to produce orthogonal poison patterns.

## S2. Main results: Clean Test Accuracy
The paper includes experiments on CIFAR-10, CIFAR-100, and ImageNet, comparing the proposed D-VAE approach with a number baselines, defending against a variety of attack types. There also additional results on other model architectures and with larger bounds. These results are pretty thorough, and the proposed approach consistently outperforms the baselines, sometimes by fairly large margins.

## S3. Ablation Study
The ablation study on the two-stage procedure for this method is good for illustrating that both stages are important for purification.

**Weaknesses:**

# W1. Motivation, necessity
a) Viability: How likely is it for the attacker to get enough poisoned samples into the training data? What percentage of the training data needs to be poisoned for their attack to succeed? Section 5.1 seems to imply that the main results poisoned *all* of the data, which is an extremely unrealistic assumption. In most scenarios, even one fifth of the data, which is the lower limit of the experiments in Table 7+8 is unrealistic.\
b) If the availability poisoning attack succeeds, then the trained model will very obviously have low performance, as can be seen in the papers. It would seem this is enough to alert the model trainer that something is wrong, leading to an inspection of the data (see below).\
c) Can poisoned data be detected more simply? For example, if there’s a loss spike, simply toss out the sample rather than try to reconstruct the original image. Particularly for large foundation models (which seem to be the most vulnerable to inadvertently incorporating a poisoned sample), recent trends toward focusing on higher quality data and common practices of checkpoint restarts when losses spike would seem to defeat this attack already, without requiring a method like the one proposed here.

# W2. Computation requirements
This method requires running a D-VAE on every sample, since it’s not known ahead of time which samples are poisoned. This would require a not insignificant amount of computation, especially in the large model setting, which is where ingesting large amounts of online data is most likely. In particular, in Section 3.5, it is proposed (rather heuristically) to run D-VAE twice, which makes it even more expensive. This also requires additional hyperparameter tuning, to tune both KLD limits, which from the “Model Training” paragraph, seem to vary between datasets. In comparison, JPEG compression is more or less “free”, as images are commonly stored in a compressed format anyway.

# W3. Technical notation
The technical notation used throughout this paper doesn’t meet the rigor expected of an academic publication. Numerous terms are used without proper introduction and are sometimes described confusingly, and there are a number of naming collisions and entities with multiple variables defined:
- Eq 3: ${kld}_{limit}$ isn’t defined.
- Eq 3: $z$ isn’t defined in the text.
- Eq 4: x_c was previously used to denote clean data, but here it’s being described as a predictive feature. Also, the subscript on x was previously used to denote the sample index i.
- Eq 4: $x = (x_c , x^t_s)$ <= The right side is later described as being predictive and non-predictive features, but $x$ is the input in the image space. Why are they being introduced as a paired element? Also, what is $t$?
- $D$ overloaded: In Figure 1 and Algorithm, $D$ denotes the decoder. In Proposition 2, D is the dimension of the Gaussian. In Equation, we have $D_{KL}$ for KL-Divergence (in contrast to KLD in Eq 3+14). In Section 3.5, D is used to refer to a poisoned dataset, as well as individual images that the input is subtracted from.
- What is the notation for the clean and poisoned dataset? In Equation 1, it appears to be $\mathcal D$ and $\mathcal T$, but in Section 4.2 it’s $C$ and $D_p$. Algorithm 1 also uses its own notation.

# W4. Writing
I encourage the authors revisit the writing. Many sections don’t read very smoothly, and I oftentimes had to re-read paragraphs multiple times to parse what was being said. Some of this has to do with a number of small grammatical or idiomatic errors, or typos (see Miscellaneous for a few non-exhuastive examples), but more of it had to do with the way concepts are introduced. For example, it’s not clear what part of the broad landscape of adversarial examples field this paper is going to be on until pretty deep into the Introduction, and my understanding of whether this paper was discussing a malicious method flipped back and forth multiple times.

## Miscellaneous:
- Title, throughout the paper: The regular use of “Poisons” throughout this paper is strange to me. “Poison” generically refers to a substance, and not something countable or discrete, as data samples are. Terms like “Poison attacks” to refer to the methodologies or “Poisoned samples” to refer to individual perturbed samples makes more sense.
- pg 1: Datasets like CIFAR-10, CIFAR-100, and ImageNet should be cited the first time they’re introduced.
- pg 2: “unlearnale”
- Figure 2: “Loss more information” <= “Lose”
- Figure 2: It’s not clear what Figure 2a is showing. The 3 methods shown haven’t been introduced, as far as I can tell, none of the lines necessarily correspond with “the paper’s method”.
- pg 4: “obtain the restored learnable tainted samples” <= Isn’t the goal to reconstruct the underlying image so it’s no longer “tainted”? The images already come “tainted”.
- Section 4.2: $\hat{D}_p$ appears multiple times with the $p$ not being subscripted.
- Table 1: “In fact, it manages to achieve an even superior attacking performance than the original poisoned dataset in most instances with less amplitude.” <= This info doesn’t seem to present in Table 1, or anywhere else in Section 4.2?
- Appendix Table 11: not centered
- Appendix Fig 4: Adversarial patterns are inherently designed to be close to imperceptible, which makes it really challenging to tell the difference between the bottom 3 rows. Is it possible to show some sort of differential instead?

**Questions:**

Q1. Instead of a VAE, how well does a non-variational auto-encoder work? Such models still have an information bottleneck, and are also trained with reconstruction loss.\
Q2. Why separately decode the perturbation and the reconstructed image? If the input $x$ is $x_c + p$, then knowing either $x_c$ or $p$ should yield the other, since $x$ is given.\
Q3. What percent of the data is poisoned in the main experiments in Tables 2-5?\
Q4. For the Gray and JPEG baselines, are similar transformations applied to the test images during evaluation? Otherwise, these approaches are likely at a disadvantage due to a domain gap.

**Details Of Ethics Concerns:**

Answer to the above question is yes and no.

This paper has to do with adversarial examples, which is a topic that has implications for safety and privacy, and clear applications in harmful applications. In particular, this paper concerns removing availability poison patterns from data. These availability attacks can be used for sabotaging model training, but can also be used to protect oneself from having one’s data being used to train a model without permission. As such, the proposed method can be used for both positive (neutralizing malicious data in a training set) and negative (circumventing someone’s attempt at data privacy) purposes.

That said, both attacks and defenses are common the adversarial research, and it doesn’t appear that this paper is introducing any ethical concerns beyond those inherent to the field. Regardless, the authors should consider including some discussion of the broader implications of their work.

---

> ### Author Response · Authors · 2023-11-16
> **Official Comments by Authors (Part 1)**
>
> Thank you for the valuable suggestions! I have submitted a revised draft addressing the comments provided by the reviewers. Below, you'll find the responses to the questions:
>
> **Q1.** Questions regarding the disentangled poison patterns in the **Strengths** part.
>
> **A1.** Given that the defender lacks groundtruth values for the perturbations $\pmb{p}$, it is not possible to optimize $\pmb{u_y}$ and $D_{\theta_p}$ to learn to predict $\pmb{\hat{p}}$ directly by minimizing ${\lVert \pmb{p} - \pmb{\hat{p}} \rVert}_2^2$ during model training. Instead, as the residuals $\pmb{x} - \pmb{\hat{x}}$ contain the majority of the groundtruth $\pmb{p}$ when imposing a low upper bound on the KLD loss, we propose minimizing ${\lVert (\pmb{x} - \pmb{\hat{x}}) - \pmb{\hat{p}} \rVert}_2^2$. As $\pmb{\hat{p}}$ is generated by $\pmb{u_y}+\pmb{z}$, which has an information bottleneck, it is hard to achieve a perfect reconstruction of $\pmb{p}$, and $\pmb{\hat{p}}$ is most likely to be a part of $\pmb{p}$. In the table below, we offer the $\ell_2$-norm of both $\pmb{p}$ and $\pmb{\hat{p}}$ , and we can see that the $\pmb{\hat{p}}$ has a smaller amplitude. In Section 4.2, the experiments show that the $\pmb{\hat{p}}$ remains effective as poison patterns. Notably, the amplitude of $\pmb{\hat{p}}$ is comparable to that of $\pmb{{p}}$, with $\pmb{\hat{p}}$ being slightly smaller than $\pmb{{p}}$ except for OPS.
> | Datasets  |                 |  EM  | REM  | NTGA | LSP  |  AR  | OPS  |
> | :-------: | :-------------: | :--: | :--: | :--: | :--: | :--: | :--: |
> | CIFAR-10  |    $\pmb{p}$    | 1.53 | 1.80 | 2.96 | 0.99 | 0.98 | 1.27 |
> |           | $\pmb{\hat{p}}$ | 1.24 | 0.92 | 0.71 | 0.73 | 0.68 | 1.77 |
> | CIFAR-100 |    $\pmb{p}$    | 1.34 | 1.73 |  -   | 0.99 | 0.98 | 1.34 |
> |           | $\pmb{\hat{p}}$ | 0.81 | 0.69 |  -   | 0.69 | 0.78 | 1.82 |
>
> Visual results of the normalized perturbations are in Figure 1, and we observe a striking similarity between $\pmb{\hat{p}}$ and $\pmb{{p}}$, especially for LSP and OPS. Since LSP and OPS use class-wise perturbations (*i.e.,* perturbations are identical for each class of images), they exhibit lower class-conditioned entropy compared to other attack methods that employ sample-wise perturbations. This makes the reconstruction of LSP and OPS perturbations much easier.
>
> **Q2.** **Ratio of Poisoned Data is Unpractical**: it is not realistic to poison the whole training data.
>
> **A2.** Our paper mainly addresses defenses against perturbative availability poisoning attacks. We have revised the "Data Poisoning" section in the related works to explicitly outline the attack setting and distinguish it from other related attacks, *e.g.,* backdoor attacks, and some types of poisoning attacks.
>
> Availability attacks aim to degrade the overall performance on validation and test datasets by solely poisoning the training dataset. Typically, such attacks inject poisoned data into the clean training set. Poisoned samples are usually generated by adding unbounded perturbations, and take only a fraction of the entire dataset [2]. The performance degradation induced by these attacks is relatively not high at a small poison ratio (*e.g.,* a 16\% drop on CIFAR-10 [3]), and the poisoned samples are relatively distinguishable [3].
>
> Another recent emerging type is perturbative availability poisoning attacks [1], where samples from the entire training dataset undergo subtle modifications (*e.g.,* bounded perturbations $\Vert \pmb{p} \Vert_{\infty} \le \frac{8}{255}$), and are correctly labeled. This type of attack, also known as unlearnable samples [4], can be viewed as a promising approach for data protection. Models trained on such datasets often approach random guessing performance on clean test data.
>
> Our paper mainly focuses on defenses against perturbative availability poisoning attacks, since defense against these attacks remains a challenging problem, *e.g.,* existing data sanitization techniques are not applicable as the entire dataset is perturbed. Therefore, unless explicitly stated otherwise, the poisoning rate in our paper is set to 100%. This choice allows us to showcase the effectiveness of the proposed defense method. In line with previous researches on perturbative availability poisoning attacks [3], we also showcase the effectiveness of our proposed methods in partial poisoning settings, *i.e.,* taking a lower poisoning rate.
>
> [1] Zhuoran Liu, Zhengyu Zhao, and Martha Larson. **Image shortcut squeezing: Countering perturbative availability poisons with compression.** In *ICML*, 2023
>
> [2] Pang Wei Koh and Percy Liang. **Understanding black-box predictions via influence functions.** In *ICML*, 2017
>
> [3] Yiwei Lu, Gautam Kamath, and Yaoliang Yu. **Exploring the limits of model-targeted indiscriminate data poisoning attacks.** In *ICML*, 2023.
>
> [4] Hanxun Huang, Xingjun Ma, Sarah Monazam Erfani, James Bailey, and Yisen Wang. **Unlearnable examples: Making personal data unexploitable.** In *ICLR*, 2021.

---

> ### Author Response · Authors · 2023-11-16
> **Official Comments by Authors (Part 2)**
>
> **Q3.** **Model Trainer Inspection of Training Data:** the model trainer can inspect training data as the model performs worse on the test data.
>
> **A3.** In certain vertical domain, the attacker is capable of poisoning the entire training dataset. The model trainer will lack clean data for validation and might remain unaware of the poisoning, rendering any inspection of the data futile.
>
>
>
> **Q4.** **Detection of Poisoned Data:** Can the poisoned data be detected more simply?
>
> **A4.** In the case of perturbative availability poisoning attacks, where the entire training dataset is poisoned, there is limited research on detecting poisoned data compared to backdoor attacks. Perturbative availability poisoning attacks also exists on attacking large foundation models, such as Anti-Dreambooth [5] that also poisons all the training samples used for fine-tuning the diffusion models. In section 5, we also demonstrate that our proposed method can be utilized to detect poisoned data, a aspect not previously explored in related research that works on defenses [1].
>
>
>
> **Q5.** Computation requirements for the purification process, and additional requirements for hyperparameters. Comparison with JPEG compression.
>
> **A5.** The table below presents the training time for D-VAE, the inference time for the poisoned dataset, and the time to train a classifier using the purified dataset. For comparison, we include the training-time defense **Adversarial Training**. It's important to note that the times are recorded using CIFAR-10 as the dataset, PyTorch as the platform, and a single Nvidia RTX 3090 as the GPU. As can see from the results, the total purification time is approximately one and a half times longer than training a classifier. Compared to adversarial training, our methods are about 5 times faster. Additionally, our method achieves an average performance around 90%, which is 15% higher than the performance achieved by adversarial training.
>
> |      Component       | Train D-VAE for twice | Perform inference on the poisoned data three times | Train a classifier | Total Time  |
> | :------------------: | :-------------------: | :------------------------------------------------: | :----------------: | :---------: |
> |      Our method      |      23 minutes       |                less than 2 minutes                 |     16 minutes     | 41 minutes  |
> | Adversarial Training |         N.A.          |                        N.A.                        |    229 minutes     | 229 minutes |
>
> For the second stage, which involves training D-VAE, the hyperparameters, including $kld_{limit}$ and downsample size, are consistently set to the same values for all datasets (CIFAR-10, CIFAR-100, and ImageNet-subset) across various attack methods and perturbation levels. While in the first stage which aims to recover most parts of poison patterns, the hyperparameters differs mostly due to different input size. Since ImageNet-subset comprises larger images (size $224 \times 224$) compared to CIFAR-10, to enhance the disentanglement of poison patterns, we employ a deeper downsampling in the experiments on ImageNet-subset.
>
> Despite JPEG compression being a more efficient defense method, its performance is lower than our methods, with an 18% lower on CIFAR-10 and a 9% lower on CIFAR-100 and ImageNet-subset. Considering the time taken for the standard training of the classifier, the total time required by our methods is about two and a half times that of JPEG, which is still within an acceptable range.
>
>
>
> **Q6.** Technical notations.
>
> **A6.** We have now fixed all the notations and standardized all the symbols in the updated manuscript.
>
> - $kld_{limit}$ is the upper bound on the KLD loss.
> - $\pmb{z}$ is the latent features from the encoder in VAE/D-VAE, which is resampled from $\mathcal{N}(\mu,\sigma)$, and $\mu,\sigma$ are the direct outputs from the encoder.
>
> - $\pmb{x_c}$ is the clean data, $\pmb{p}$ is the added perturbations, $\pmb{x}$ is the poisoned data, $y$ is the label.
> - $\mathcal{D}$ is the clean test dataset, $\mathcal{T}$ is the clean training dataset, $\mathcal{P}$ is the poisoned training dataset.
> - $\pmb{\hat{x}}$ is the reconstructed data by VAEs/D-VAEs, $\pmb{\hat{p}}$ is the disentangled perturbations by D-VAEs, $\pmb{z}$ is the latents encoded by the VAEs/D-VAEs.
> - D-VAE consists of encoder $E_{\phi}$, decoder $D_{\theta_c}$, auxiliary decoder $D_{\theta_p}$, and class-wise embeddings ${\pmb{u_y}}$, which are all trainable.
> - For the theoretical analysis in Section 3.3, we formulate expressions in the feature space, and thus we now change $\pmb{x}$ to $\pmb{v}$ to denote the features. And $\pmb{v}=(\pmb{v_c},\pmb{v_s^t})$ denote the paired features extracted from the data. $t$ is used to denote that the feature is non-predictive.
>
> [5] Thanh Van Le, Hao Phung, Thuan Hoang Nguyen, Quan Dao, Ngoc Tran and Anh Tran. **Anti-DreamBooth: Protecting Users from Personalized Text-to-Image Synthesis.** In *ICCV*, 2023.

---

> ### Author Response · Authors · 2023-11-16
> **Official Comments by Authors (Part 3)**
>
> **Q7.** Concerns regarding writings.
>
> **A7.** Thank you once again for your thorough review of our paper. I have revisited the content and addressed all the mentioned concerns. The revised sections are highlighted in blue in the resubmitted draft.
>
>
>
> **Q8.** Instead of a VAE, how does a non-variational auto-encoder with an information bottleneck work?
>
> **A8.** In the newly added Appendix E, we conduct experiments on purification using non-variational auto-encoders (AEs) with an information bottleneck. To achieve non-variational auto-encoders with different bottleneck levels, we modify the width of the features within the auto-encoder architecture. This results in models with varying parameter numbers. Then, we proceed to train the AE on the poisoned CIFAR-10 dataset, and test on the clean test dataset with classifiers trained on the purified dataset.
> As illustrated in Figure 5 in the Appendix, a non-variational auto-encoder with an information bottleneck can also eliminate poison patterns. However, when comparing at a similar level of reconstruction quality measured by PSNR, VAEs demonstrate a stronger inclination to remove more poison patterns in both the REM and LSP poisoning attacks. For EM poisoning attacks, the outcomes are comparable. These observations align with the theoretical analysis presented in Section 3.3.
>
>
>
> **Q9.** Why separately decode the perturbation and the reconstructed image? If the input $\pmb{x}$ is $\pmb{x_c}+\pmb{p}$, then knowing either $\pmb{x_c}$ or $\pmb{p}$ should yield the other, since $\pmb{x}$ is given.
>
> **A9.** As observed in Section 3.2, setting a low upper bound on the KLD loss results in the reconstructed images $\pmb{\hat{x}}$ containing very few poison patterns but being heavily corrupted in terms of PSNR (*e.g.,* PSNR is around 22 when the upper bound on the KLD loss is set to 1.0). Consequently, the residuals $\pmb{x} - \pmb{\hat{x}}$ contains both the majority of poison patterns and some parts of $\pmb{x_c}$ that cannot be reconstructed due to the information bottleneck in the latents $\pmb{z}$. Therefore, with an amplitude around 18 in terms of $\ell_2$-norm for $\pmb{x} - \pmb{\hat{x}}$ and 1.5 for $\pmb{p}$, attempting to estimate $\pmb{p}$ based on $\pmb{x} - \pmb{\hat{x}}$ will result in a significant error.
>
> At the same time, since the decoder for generating the disentangled perturbations $\pmb{\hat{p}}$ is trained in an unsupervised manner by minimizing ${\lVert (\pmb{x} - \pmb{\hat{x}}) - \pmb{\hat{p}} \rVert}_2^2$. Considering the bottleneck in $\pmb{z} + \pmb{u_y}$, $\pmb{\hat{p}}$ cannot be fully reconstructed and typically represents only parts of the groundtruth perturbations $\pmb{p}$. Consequently, although $\pmb{x}-\pmb{\hat{p}}=\pmb{x_c}+(\pmb{p}-\pmb{\hat{p}})$ contains fewer poison patterns than $\pmb{x}$, the remaining patterns can still be effective for poisoning.
>
> Based on the aforementioned concept, we employ a two-stage purification framework in Section 3.5. In the first stage, we utilize a small KLD limit and retain $\pmb{x}-\pmb{\hat{p}}$, containing fewer poison patterns. In the second stage, a large KLD limit is applied, preserving $\pmb{\hat{x}}$, which exhibits good reconstruction quality and retains most useful features for classification.
>
>
>
> **Q10.** What percent of the data is poisoned in the main experiments in Tables 2-5?
>
> **A10.** Our paper mainly addresses defenses against perturbative availability poisoning attacks, where the entire training data is poisoned with bounded perturbations. Consequently, in the main experiments presented in Tables 2-5, the poisoning rate is consistently set to 100%.

---

> ### Author Response · Authors · 2023-11-16
> **Official Comments by Authors (Part 4)**
>
> **Q11.** For the Gray and JPEG baselines, are similar transformations applied to the test images during evaluation? Otherwise, these approaches are likely at a disadvantage due to a domain gap.
>
> **A11.** For the Grayscale, JPEG, and BDR baselines, we adhere to the original settings in ISS [1], applying these transformations only to the training images and not to the test images during evaluation. Here, we report the performance when the same transformations are applied to the test images in the table below. Note that we select the CIFAR-10 as the dataset. From the results in the table, we observe that adopting the same transformations to the test images does not significantly affect the performance, and our method consistently outperforms the baselines.
>
> | Attacks/Defenses | BDR (Train) | Gray (Train) | JPEG (Train) | BDR (Both) | Gray (Both) | JPEG (Both) | **Ours**  |
> | :--------------: | :---------: | :----------: | :----------: | :--------: | :---------: | :---------: | :-------: |
> |       NTGA       |    57.80    |    65.26     |    78.97     |   62.57    |    65.51    |    70.98    | **89.21** |
> |        EM        |    81.91    |    19.50     |    85.61     |   84.77    |    19.45    |    79.61    | **91.42** |
> |       TAP        |    80.18    |    21.50     |    84.99     |   80.62    |    21.47    |    79.46    | **90.48** |
> |       REM        |    32.36    |    62.35     |    84.40     |   28.58    |    62.31    |    78.49    | **86.38** |
> |       SEP        |    81.21    |     8.47     |    84.97     |   79.27    |    8.36     |    79.54    | **90.74** |
> |       LSP        |    40.25    |    73.63     |    79.91     |   48.43    |    73.61    |    69.56    | **91.20** |
> |        AR        |    29.14    |    36.18     |    84.97     |   45.02    |    35.93    |    79.41    | **91.77** |
> |       OPS        |    19.58    |    19.43     |    77.33     |   18.49    |    19.15    |    69.45    | **88.96** |
> |       Mean       |    52.80    |    38.29     |    78.89     |   55.97    |    38.22    |    65.88    | **90.02** |
>
>
>
> **Q12.** Ethics concerns.
>
> **A12.** Thank you for the guidance. We have incorporated an **ETHICS STATEMENT** section following the main content of the paper. We also list the content here.
>
> In summary, our paper presents a effective defense strategy against perturbative availability poisoning attacks, which aim to undermine the overall performance on validation and test datasets by introducing imperceptible perturbations to training examples with accurate labels. Perturbative availability poisoning attacks are viewed as a promising avenue for data protection, particularly to thwart unauthorized use of data that may contain proprietary or sensitive information. However, these protective methods pose challenges to data exploiters who may interpret them as potential threats to a company's commercial interests. Consequently, our method can be employed for both positive usage, such as neutralizing malicious data within a training set, and negative purpose, including thwarting attempts at preserving data privacy. Our proposed method not only serves as a powerful defense mechanism but also holds the potential to be a benchmark for evaluating existing attack methods. We believe that our paper contributes to raising awareness about the vulnerability of current data protection techniques employing perturbative availability poisoning attacks. This, in turn, should stimulate further research towards developing more reliable and trustworthy data protection techniques.

---

> ### Comment · Reviewer_D3kf · 2023-11-20
>
> Thanks authors, for the response. I’ve read them, along with the other reviews.
>
> > Disentangled poison patterns.
>
> I’m still a little confused why we want to minimize $||(x-\hat{x}) - \hat{p}||_2^2$. If we learn this perfectly, then we predict a $\hat{p}$ that we could have gotten simply by subtracting our cleaned prediction $\hat{x}$ from the input $x$. I also stand by my original comments that the reconstructed poisons are not actually that close, and not what I would describe as “striking”. Amplitude is not enough for comparing the noise patterns. What is the reconstruction error, relative to the noise amplitude? Or PSNR?
>
> > Ratio of Poisoned Data is Unpractical
>
> I’m still not convinced on this point. In the scenario that is described in the second paragraph of the Introduction (scraping data from the Internet), it seems incredibly unlikely that all the data that one would scrape would be poisoned. The only scenario that I can currently envision then is if someone or some entity has specifically poisoned a dataset (almost as a form of encryption), and we then try to train a model on it. In that case, a) why would the poisoned dataset be available to us in the first place, and b) why would want to train on it? Regardless, this seems like a situation where we shouldn’t be trying to train on the poisoned dataset in the first place, as we’re clearly going against the dataset owner’s wishes, or training on something that should be recognizable as something designed to produce garbage results--in which case we should just go find other data.
>
> > Computation
>
> Thank you for providing this. I encourage the authors to include this discussion in their paper. I somewhat agree with the authors that data purification taking 1.5x longer than training may be acceptable, especially since this should hopefully only have to be done once, but this is still a disadvantage worth acknowledging, especially since the JPEG approach is already “baked in” to virtually every training pipeline and does provide competitive protection.
>
> > Ethics Statement
>
> Thanks for adding this to the paper.

---

> > ### Author Response · Authors · 2023-11-20
> > **Official Comment by Authors (Part 1)**
> >
> > Thank you for the valuable suggestions! I have submitted a revised draft adding discussion on computation and comparisons with JPEG in Appendix F. Below, you'll find the responses to the questions:
> >
> > **Q1.** Disentangled poison patterns.
> >
> > **A1.** We acknowledge that minimizing ${\lVert \pmb{p} - \pmb{\hat{p}} \rVert}_2^2$ during model training would likely result in superior reconstruction outcomes. However, in this paper, we operate under the assumption that the defender or model trainer lacks access to the groundtruth values of $\pmb{p}$, which is a more practical scenario. Consequently, we propose the minimization of ${\lVert (\pmb{x} - \pmb{\hat{x}}) - \pmb{\hat{p}} \rVert}_2^2$. Therefore, even though this form of minimization may not achieve perfect reconstruction, our experiments indicate that incorporating such reconstruction in the proposed method significantly contributes to performance.
> >
> > I agree that the similarity between $\pmb{p}$ and $\pmb{\hat{p}}$ is not striking, and they are more likely to have a similar attacking function as shown in Section 4.2 of the main paper. In the table below, we present quantitative measures of the reconstruction error between $\pmb{p}$ and $\pmb{\hat{p}}$ in terms of MSE and PSNR. Although the reconstruction may not be perfect in terms of the reconstruction error, the main advantage of disentangling perturbations is evident in the effective removal of the majority of poison patterns in the first stage. This facilitation contributes to generating poison-free purified samples in the second stage. To further emphasize this point, in Section 4.4 of the main paper, we conduct an ablation study. As shown in Table 6 in the main paper, the removal of poison patterns in the first stage significantly influences performance, leading to an almost 10% increase.
> >
> > | Datasets  |     Metrics     |  EM  | REM  | NTGA | LSP  |  AR  | OPS  |
> > | :-------: | :-------------: | :--: | :--: | :--: | :--: | :--: | :--: |
> > | CIFAR-10  | MSE ($10^{-4}$) | 4.4  | 6.6  |  12  | 3.5  | 5.2  | 2.3  |
> > |           |    PSNR (dB)    | 33.6 | 31.8 | 29.3 | 34.7 | 32.9 | 36.5 |
> > | CIFAR-100 | MSE ($10^{-4}$) | 4.1  | 7.1  |  -   | 4.1  | 4.7  | 2.0  |
> > |           |    PSNR (dB)    | 33.9 | 31.5 |  -   | 34.0 | 33.1 | 37.3 |
> >
> > Additionally, we present the reconstruction error in terms of MSE and PSNR when utilizing $\pmb{x} - \pmb{\hat{x}}$ to estimate the perturbations $\pmb{p}$. As depicted in the table below, estimating with $\pmb{x} - \pmb{\hat{x}}$ leads to a significantly larger error.
> >
> > | Datasets  |     Metrics     |  EM  | REM  | NTGA | LSP  |  AR  | OPS  |
> > | :-------: | :-------------: | :--: | :--: | :--: | :--: | :--: | :--: |
> > | CIFAR-10  | MSE ($10^{-4}$) | 55.8 | 54.9 | 52.3 | 55.4 | 60.4 | 55.6 |
> > |           |    PSNR (dB)    | 22.7 | 22.8 | 22.8 | 22.8 | 22.3 | 22.7 |
> > | CIFAR-100 | MSE ($10^{-4}$) | 53.8 | 55.3 |  -   | 55.7 | 60.6 | 55.4 |
> > |           |    PSNR (dB)    | 22.9 | 22.8 |  -   | 22.7 | 22.2 | 22.8 |

---

> > ### Author Response · Authors · 2023-11-20
> > **Official Comment by Authors (Part 2)**
> >
> > **Q2.** Ratio of Poisoned Data is Unpractical.
> >
> > **A2.** Perturbative availability poisoning attacks, often referred to as unlearnable examples [2], are proposed to serve as a data protection approach to prevent unauthorized training/exploration on the data. In an extreme scenario, an individual, such as a data publisher, may aim to render the entire dataset unlearnable for training DNNs while ensuring that the released data maintains high visual quality. Therefore, if the model trainer aims to train a useful model on such a dataset, they must choose a better training algorithm or seek an alternative usable dataset. In this case, the goal of the data publisher is fulfilled. Additionally, in areas where data is scarce or not released due to privacy concerns, finding another useful dataset can be challenging.
> >
> > A more common scenario is that an individual only needs to add the noise to his/her own part of the data to make it unlearnable, for example, a person adds perturbations to all his/her photos to prevent unauthorized learning of the personal information, such as identity. Essentially, even though not explicitly discussed in our paper, researches [2,3,4] reveal that models trained using a combination of poisoned and clean data are nearly equivalent to models trained solely on the clean data. Consequently, the training process gains little valuable knowledge from the poisoned data, which is why we refer to it as unlearnable examples in some cases.
> >
> > We acknowledge that poisoning the entire dataset represents an extreme scenario. Consistent with prior research emphasizing more robust attacks [2,3,4] or proposing for more potent defenses [1], we illustrate the effectiveness of our proposed methods against such attacks in the case that the entire dataset is poisoned. The primary experiments in this paper focus on an extreme scenario where the attack is most potent. Additionally, we conduct experiments in a more common scenario where only a portion of the dataset is poisoned.
> >
> >
> > [1] Zhuoran Liu, Zhengyu Zhao, and Martha Larson. **Image shortcut squeezing: Countering perturbative availability poisons with compression.** In *ICML*, 2023
> >
> > [2] Hanxun Huang, Xingjun Ma, Sarah Monazam Erfani, James Bailey, and Yisen Wang. **Unlearnable examples: Making personal data unexploitable.** In *ICLR*, 2021.
> >
> > [3] Pedro Sandoval-Segura, Vasu Singla, Jonas Geiping, Micah Goldblum, Tom Goldstein, and David Jacobs. **Autoregressive perturbations for data poisoning.** In *NeurIPS*, 2022.
> >
> > [4] Rui Wen, Zhengyu Zhao, Zhuoran Liu, Michael Backes, Tianhao Wang, and Yang Zhang. **Is adversarial training really a silver bullet for mitigating data poisoning?** In *ICLR*, 2023.

---

> ### Author Response · Authors · 2023-11-20
> **Official Comment by Authors (Part 3)**
>
> **Q3.** Computation and comparison with JPEG.
>
> **A3.** Thank you for the suggestions. We add a discussion on computation and a comparison with JPEG in the Appendix F. We also note a limitation in the JPEG compression approach used in ISS [1]—specifically, they set the JPEG quality to 10 to purify poisoned samples, resulting in significant image degradation. In the table below, we present results using JPEG with various quality settings. We also include the PSNR between purified images and input images in the table. Notably, our proposed methods consistently outperform JPEG compression when applied at a similar level of image corruption.
>
> | Attacks/Defenses | JPEG (quality 10) PSNR 22 | JPEG (quality 30) PSNR 25 | JPEG (quality 50) PSNR 27 | JPEG (quality 70) PSNR 28 | **Ours** PSNR 28 |
> | :--------------: | :-----------------------: | :-----------------------: | :-----------------------: | :-----------------------: | :--------------: |
> |       NTGA       |           78.97           |           66.83           |           64.28           |           60.19           |    **89.21**     |
> |        EM        |           85.61           |           70.48           |           54.22           |           42.23           |    **91.42**     |
> |       TAP        |           84.99           |           84.82           |           77.98           |           57.45           |    **90.48**     |
> |       REM        |           84.40           |           77.73           |           71.19           |           63.39           |    **86.38**     |
> |       SEP        |           84.97           |           87.57           |           82.25           |           59.09           |    **90.74**     |
> |       LSP        |           79.91           |           42.11           |           33.99           |           29.19           |    **91.20**     |
> |        AR        |           84.97           |           89.17           |           86.11           |           80.01           |    **91.77**     |
> |       OPS        |           77.33           |           79.01           |           68.68           |           59.81           |    **88.96**     |
> |       Mean       |           78.89           |           74.71           |           67.33           |           56.42           |    **90.02**     |
>
> As evident from the observations in Section 3.2 and the theoretical analysis in Section 3.3 of the main paper, the effectiveness of JPEG compression may be largely due to its severe degradation on the poisoned images, while VAEs demonstrate a tendency to remove poison patterns. Therefore, JPEG may exhibit suboptimal performance in the presence of larger perturbation bounds. In contrast, our method excels in eliminating the majority of poison patterns in the first stage, rendering it more robust to larger perturbation bounds. Table 5 in Section 4.3 of the main paper illustrates that when confronted with LSP attacks with larger bounds, our method demonstrates significantly smaller performance degradation compared to JPEG (with quality 10), *e.g.,* 86.13% Vs. 41.41% in terms of test accuracy.

---

> ### Comment · Reviewer_D3kf · 2023-11-22
>
> Thank you for the additional responses and updates to the paper.
>
> I understood that $\hat{p}$ is not available as an optimization target, but the proposed alternative is not completely satisfactory either. $\hat{p}$ should be directly producible from a correct cleaned image prediction $\hat{x}$, and because both $\hat{x}$ and $\hat{p}$ are model outputs, it's hard to say how optimization will choose to minimize this objective: a sizable component of the perturbation may end up being learned into $\hat{x}$.
>
> Regardless, due to the other clarifications and improvements, I may be willing to upgrade my score to 6, but I still feel this paper is somewhat borderline due to the overall premise and methodology.

---

> > ### Author Response · Authors · 2023-11-22
> > **Official Comments by Authors**
> >
> > Thank you for the valuable suggestions! I have submitted a revised draft adding additional experiments in Appendix G by training D-VAE on all attack methods with various upper bounds for the KLD loss. Below, you'll find the responses to the questions:
> >
> > **Q1.** A sizable component of the perturbation may end up being learned into $\pmb{\hat{x}}$.
> >
> > **A1.** We acknowledge that your concerns may arise in certain cases, such as when the upper bound on the KLD loss is not set low. Nevertheless, when the KLD loss is set to a low value, the presence of poison patterns in the reconstructed $\pmb{\hat{x}}$ is shown to be minimal. This observation is supported by both empirical experiments in Section 3.2 and theoretical explanations provided in Section 3.3. These outcomes are primarily attributed to the fact that the reconstruction of $\pmb{\hat{x}}$ depends on the information encoded in the latent representation $\pmb{z}$, *i.e.,* $\pmb{\hat{x}}$ is directly generated from $\pmb{z}$ using a decoder. The theoretical insights discussed in Section 3.3 highlight that **Theorem 1** indicates that perturbations which create strong attacks tend to have a larger inter-class distance and a smaller intra-class variance. Additionally, **Theorem 2** and **Remark 1** indicate that poison patterns possessing these characteristics are more likely to be  eliminated when aligning the features with a normal Gaussian  distribution (as done by the VAE).
> >
> > To further validate these observations, we now include additional experiments in Appendix G by training D-VAE on all attack methods with various upper bounds for the KLD loss. Additionally, we have performed experiments on attacks with larger perturbations. Notably, we have added results on the clean dataset for comparison. As depicted in Figure 6 in the Appendix, when the upper bound on the KLD loss is set below 1.0, the curves of the results on the poisoned dataset align closely with the results on the clean dataset. Furthermore, as the upper bound decreases, the removal of poison patterns in the reconstructed $\pmb{\hat{x}}$ increases. While it is evident that larger perturbations may be better retained in $\pmb{\hat{x}}$, it is a cat-and-mouse game between defense and attack. Additionally, larger perturbations tend to be more noticeable. These findings affirm that the observations hold for all existing attack methods, and setting a low upper bound (*e.g.,* 1.0, as in the main experiments) on the KLD loss significantly ensures that $\pmb{\hat{x}}$ contains few poison patterns. This is why we choose a low upper bound in the first stage and propose to eliminate the disentangled poison patterns.
> >
> > Defending against perturbative poisoning attacks is a challenging problem, and we believe that our proposed method can serve as a benchmark to evaluate the effectiveness of existing attacks. It can also **contribute to research efforts aimed at developing more reliable methods for robust data protection, that prevents unauthorized training, and exploration of data**.

---

> > ### Author Response · Authors · 2023-11-23
> >
> > Dear Reviewer D3kf,
> >
> > Thanks again for your great efforts and insightful comments in reviewing this paper! With the discussion period drawing to a close, we expect your feedback and thoughts on our reply. We look forward to hearing from you, and we can further address unclear explanations and remaining concerns if any.
> >
> > Best regards,
> >
> > The authors

---

### Official Review · Reviewer_kPQt · 2023-11-01

**Soundness:** 3 good
**Presentation:** 1 poor
**Contribution:** 3 good
**Rating:** 5
**Confidence:** 2

**Summary:**

This paper aims to build a purification framework to defend against data poisoning attacks, by using the proposed disentangle variational autoencoder (D-VAE). The experiments show that it achieves remarkable performance on CIFAR-10/100 and ImageNet-subset.

**Strengths:**

- The experiments show that the proposed D-VAE outperforms other state-of-the-arts.

**Weaknesses:**

My concerns are as follows.
- The math symbols should be unified. In Section 3.1, $\mathbf{x}$ represents the clean data, while in the rest of the paper, it represents the poisoned data.
- I'm really confused about the proposed method. It would be helpful if the authors could provide more information in Section 3.2. Specifically, please give a detailed description and analysis of Figure 2. The current version is hard to understand, to me.
- I'm confused about the poison recovering part in Eq. 14 and Figure 1. It seems the first two items in Eq.14 are contradictory. Are you aiming to decrease the L2 error between the reconstruction error and a tensor output by the network? I cannot understand the meaning of this term, to me.

Please clarify these points.

**Questions:**

See weaknesses.

---

> ### Author Response · Authors · 2023-11-16
> **Official Comments by Authors (1/2)**
>
> Thank you for the valuable suggestions! I have submitted a revised draft addressing the comments provided by the reviewers. Below, you'll find the responses to the questions:
>
> **Q1.** Math symbols should be unified.
>
> **A1.** We have now standardized all the symbols in the updated manuscript.
>
> - $\pmb{x_c}$ is the clean data, $\pmb{p}$ is the added perturbations, $\pmb{x}$ is the poisoned data, $y$ is the label.
> - $\mathcal{D}$ is the clean test dataset, $\mathcal{T}$ is the clean training dataset, $\mathcal{P}$ is the poisoned training dataset.
> - $\pmb{\hat{x}}$ is the reconstructed data by VAEs/D-VAEs, $\pmb{\hat{p}}$ is the disentangled perturbations by D-VAEs, $\pmb{z}$ is the latents encoded by the VAEs/D-VAEs.
> - D-VAE consists of encoder $E_{\phi}$, decoder $D_{\theta_c}$, auxiliary decoder $D_{\theta_p}$, and class-wise embeddings ${\pmb{u_y}}$, which are all trainable.
>
>
>
> **Q2.** More information in Section 3.2 and a detailed description and analysis of Figure 2.
>
> **A2.** We have revised the majority of Section 3.2, providing a comprehensive description and analysis.
>
> - Among earlier works, ISS [1] suggests the use of JPEG compression as an effective method to eliminate poisoned patterns. The central idea discussed in Section 3.2 aims to illustrate that VAEs tend to remove more poison patterns compared to JPEG compression. To investigate this, we employ EM, REM, or LSP as the attack methods to construct the poisoned dataset.
> - We proceed to train VAEs with varying upper bounds for the KLD loss on the poisoned dataset $\mathcal{P}$. Following this, we report the accuracy on the clean test set $\mathcal{D}$ achieved by a ResNet-18 trained on the reconstructed data $\pmb{\hat{x}}$. In the left image of Figure 2(a), It is observed that reducing the KLD loss leads to a decrease in reconstruction quality, measured by the PSNR between $\pmb{\hat{x}}$ and $\pmb{{x}}$. Additionally, this reduction eliminates added poison patterns and original valuable features. The right image of Figure 2(a) shows that the increased removal of poison patterns in $\pmb{\hat{x}}$ correlates with improved test accuracy. However, when $\pmb{\hat{x}}$ is heavily corrupted, further reducing the KLD loss removes more valuable features, leading to a drop in test accuracy.
> - In Figure 2(b), we contrast the outcomes of VAEs with JPEG compression at different quality settings. Across all results, when processed through VAEs and JPEG to achieve similar PSNR, the test accuracy with VAEs surpasses that of JPEG. This suggests that VAEs are significantly more effective at eliminating poison patterns than JPEG compression, when achieving similar levels of reconstruction quality.
>
>
> [1] Zhuoran Liu, Zhengyu Zhao, and Martha Larson. **Image shortcut squeezing: Countering perturbative availability poisons with compression.** *Proc. Int’l Conf. Machine Learning*, 2023

---

> ### Author Response · Authors · 2023-11-16
> **Official Comments by Authors (2/2)**
>
> **Q3.** Are the first two items in Eq. 14 to train the D-VAE contradictory?
>
> **A3.** We have revised and provided a detailed explanation of the components in Section 3.4, focusing on the learning process of disentangled perturbations. In conclusion, the two terms are not contradictory as both the reconstruction of $\pmb{\hat{x}}$ and $\pmb{\hat{p}}$ have an information bottleneck.
>
> - A comprehensive explanation of the loss design for training D-VAE.
>
>   - Given that the defender lacks groundtruth values for the perturbations $\pmb{p}$, it is not possible to optimize $\pmb{u_y}$ and $D_{\theta_p}$ to learn to predict $\pmb{\hat{p}}$ directly by minimizing ${\lVert \pmb{p} - \pmb{\hat{p}} \rVert}_2^2$ during model training.
>
>   - Expanding on the insights from Section 3.2 and **Remark 1** in Section 3.3, when imposing a low upper bound on the KLD loss, creating an information bottleneck on the latents $\pmb{z}$, the reconstructed $\pmb{\hat{x}}$ cannot achieve perfect reconstruction, making poison patterns more challenging to be recovered in $\pmb{\hat{x}}$. As a result, a significant portion of poison patterns $\pmb{p}$ persists in the residuals $\pmb{x} - \pmb{\hat{x}}$.
>
>   - Following **Proposition 2** in Section 3.3, the majority of poison patterns associated with each class data exhibit relatively low entropy, suggesting that they can be predominantly reconstructed using representations with limited capacity. Considering that most poison patterns are crafted to be sample-wise, we propose a learning approach that maps the summation of a trainable class-wise embedding $\pmb{u_y}$ and the latents $\pmb{z}$ to $\pmb{\hat{p}}$ through an auxiliary decoder $D_{\theta_p}$.
>
>   - To learn $\pmb{u_y}$ and train $D_{\theta_p}$, we propose minimizing ${\lVert (\pmb{x} - \pmb{\hat{x}}) - \pmb{\hat{p}} \rVert}_2^2$, as the residuals $\pmb{x} - \pmb{\hat{x}}$ contain the majority of the groundtruth $\pmb{p}$ when imposing a low upper bound on the KLD loss.
>
> - Next, let's analyze why ${\lVert \pmb{x} - \pmb{\hat{x}} \rVert}_2^2$ and ${\lVert (\pmb{x} - \pmb{\hat{x}}) - \pmb{\hat{p}} \rVert}_2^2$ are not contradictory.
>   - As there is a information bottleneck in the latents $\pmb{z}$ and $\pmb{u_y}$, we assume that the best-reconstructed images are represented by $\pmb{\tilde{x}}$, and the best-reconstructed perturbations are represented by $\pmb{\tilde{p}}$.
>
>   - If some parts of $\pmb{\tilde{x}}$ are not recovered in $\pmb{\hat{x}}$, we denote $\pmb{\hat{x}} = \pmb{\tilde{x}} - \pmb{\Delta{x}}$.
>
>   - As $\pmb{\hat{p}}$ is generated by $\pmb{u_y}+\pmb{z}$, the $\pmb{\Delta x}$ could be recovered in $\pmb{\hat{p}}$, leading to $\pmb{\hat{p}} = \pmb{\tilde{p}} + \pmb{\Delta{x}}$.
>
>   - We can observe that ${\lVert \pmb{x} - \pmb{\hat{x}} \rVert}_2^2 + {\lVert (\pmb{x} - \pmb{\hat{x}}) - \pmb{\hat{p}} \rVert}_2^2$ is minimized when $\pmb{{\Delta x}}= \pmb{0}$, as ${\lVert \pmb{x} - \pmb{\hat{x}} \rVert}_2^2 + {\lVert (\pmb{x} - \pmb{\hat{x}}) - \pmb{\hat{p}} \rVert}_2^2 = {\lVert \pmb{x} - \pmb{\tilde{x}} + \pmb{\Delta{x}} \rVert}_2^2 + {\lVert (\pmb{x} - \pmb{\tilde{x}}) - \pmb{\tilde{p}} \rVert}_2^2$ .
>
>   - When $\pmb{{\Delta x}}= \pmb{0}$, ${\lVert \pmb{x} - \pmb{\hat{x}} \rVert}_2^2 + {\lVert (\pmb{x} - \pmb{\hat{x}}) - \pmb{\hat{p}} \rVert}_2^2= {\lVert \pmb{x} - \pmb{\tilde{x}} \rVert}_2^2 + {\lVert (\pmb{x} - \pmb{\tilde{x}}) - \pmb{\tilde{p}} \rVert}_2^2$ .
>
>   - In the table below, we present quantitative results indicating that $\pmb{\hat{x}}$ reconstructed by VAE or D-VAE exhibits similar PSNR when the same upper bound for KLD loss is employed. From the PSNR between $\pmb{x}$ and $\pmb{\hat{x}}$+$\pmb{\hat{p}}$, we observe that $\pmb{\tilde{p}}$ can recover some parts of $\pmb{p}$, making $\pmb{\hat{x}} + \pmb{\hat{p}}$ closer to $\pmb{x}$.
>
>     |                    Model/KLD loss                    |  2.0  |  2.5  |  3.0  |  3.5  |  4.0  |
>     | :--------------------------------------------------: | :---: | :---: | :---: | :---: | :---: |
>     |          VAE: PSNR($\pmb{x},\pmb{\hat{x}}$)          | 25.19 | 26.14 | 27.02 | 27.67 | 28.27 |
>     |         D-VAE: PSNR($\pmb{x},\pmb{\hat{x}}$)         | 25.07 | 26.12 | 26.97 | 27.54 | 28.19 |
>     | D-VAE: PSNR($\pmb{x},\pmb{\hat{x}}$+$\pmb{\hat{p}}$) | 25.64 | 26.71 | 27.70 | 28.42 | 29.07 |
>
>
>
>
>
> [1] Zhuoran Liu, Zhengyu Zhao, and Martha Larson. **Image shortcut squeezing: Countering perturbative availability poisons with compression.** *Proc. Int’l Conf. Machine Learning*, 2023

---

> ### Author Response · Authors · 2023-11-21
>
> Dear Reviewer kPQt,
>
>
> Thank you for taking the time to review our submission and providing us with constructive comments and a favorable recommendation. We would like to know if our responses adequately addressed your earlier concerns. Additionally, if you have any further concerns or suggestions, we would be more than happy to address and discuss them to enhance the quality of the paper. We eagerly await your response and look forward to hearing from you.
>
> Best regards,
>
> The authors

---

### Author Response · Authors · 2023-11-16
**Summary of Modification**

We thank all reviewers for the insightful comments to improve our work, we label the modification in blue color in the revised manuscript and summarize the updates as below:

1. We have now fixed all the notations and standardized all the symbols.
2. We added a comparison with non-variational autoencoders which have an information bottleneck in Appendix E.
3. We have incorporated additional related works that cover various poisoning attacks and provided a detailed discussion on the specific focus of this paper.

4. We added a more thorough analysis of Figure 2 in Section 3.2.
5. We added a more comprehensive analysis on the loss design in Equation 14 in Section 3.2.
6. We replaced the term "poisons" with "poisoning attacks" when referring to the attacks, and "poison perturbations" when referring to the perturbations in some cases.
7. We added more visual results of the residuals in Figure 4 in the Appendix.
8. We added the **ETHICS STATEMENT** on page 10 after the main content.

---

### Meta-Review · Area_Chair_ap9E · 2023-12-12

**Metareview:**

There was a lengthy discussion among the reviewers. The rebuttal addressed some concerns, but some serous concerns are still remaining. None of the reviewers is strongly advocating accepting this paper, so it will not be accepted. The main concern is the problem setting assumption regarding poisoning the entire training set. Such an assumption has been used in prior published papers too, but the AC agrees with Reviewer D3kf that existence of such prior work does not necessitate publishing newer work using the same non-practical assumptions.

**Justification For Why Not Higher Score:**

The main concern is the problem setting assumption regarding poisoning the entire training set. Such an assumption has been used in prior published papers too, but the AC agrees with Reviewer D3kf that existence of such prior work does not necessitate publishing newer work using the same non-practical assumptions.

**Justification For Why Not Lower Score:**

N/A

---

### Decision · Program_Chairs · 2024-01-16

Reject